# Beyond Mode Collapse: Distribution Matching for Diverse Reasoning

**Xiaozhe Li**[1]  **Yang Li**[2]  **Xinyu Fang**[3 4]  **Shengyuan Ding**[3 5]  **Peiji Li**[3 5]  **Yongkang Chen**[3]  **Yichuan Ma**[3 5]
**TianYi Lyu**[1]  **Linyang Li**[3 5 6]  **Dahua Lin**[3 6]  **Qipeng Guo**[3 5]  **Qingwen Liu**[1]  **Kai Chen**[3]

## Abstract

On-policy reinforcement learning methods like GRPO suffer from *mode collapse*: they exhibit reduced solution diversity, concentrating probability mass on a single solution once discovered and ceasing exploration of alternative strategies. We show this stems from reverse KL minimization's mode-seeking behavior, which reinforces the first high-reward trajectory found rather than maintaining a distribution over multiple diverse solutions. We propose DMPO (**D**istribution-**M**atching **P**olicy **O**ptimization), which prevents mode collapse through principled approximation of forward KL minimization. DMPO constructs a group-level target distribution over sampled trajectories proportional to their rewards, then aligns the policy distribution to this target. This provides mode-covering behavior without requiring sampling from the intractable global target distribution, enabling sustained exploration throughout training. We validate DMPO on NP-hard combinatorial optimization, where exponentially many feasible solutions exist but only a few approach optimality—an ideal testbed for evaluating exploration. DMPO achieves 43.9% Quality Ratio on text-based NP-Bench (vs. GRPO's 40.1%) and 43.1% on vision-based NP-Bench (vs. 38.4%)—demonstrating 9% and 12% relative improvements respectively. These gains generalize to mathematical reasoning (+2.0%) and out-of-domain tasks (+2.3%), showing that diversity-preserving training enhances general reasoning capabilities across modalities. Our work establishes distribution matching as a practical, principled approach to preventing mode collapse in on-policy RL, with consistent quality improve-

ments demonstrating sustained exploration across diverse reasoning tasks.

## 1. Introduction

Reinforcement Learning with Verifiable Rewards (RLVR) has emerged as a powerful paradigm for training large reasoning models (LRMs), enabling substantial gains in mathematical problem solving (He et al., 2025; Yu et al., 2025) and code generation (Liu & Zhang, 2025). By replacing learned reward models with exact, rule-based verification, RLVR enables stable optimization and scalable improvement. As a result, on-policy methods such as GRPO (Shao et al., 2024a) have become standard tools for training reasoning-oriented language models.

Despite their success, these methods suffer from a fundamental limitation that becomes increasingly severe as training progresses: *mode collapse* (Karan & Du, 2025; Yue et al., 2025; Song et al., 2025). Once a single high-reward solution is discovered, the policy rapidly concentrates probability mass on that trajectory, suppressing alternative solution strategies and effectively halting exploration. This premature convergence prevents discovery of better solutions, reduces robustness, and leads to brittle reasoning behaviors that fail to generalize.

We show that this phenomenon is not an implementation artifact, but a direct consequence of the optimization objective used in standard on-policy RL. GRPO and related methods implicitly minimize the *reverse* KL divergence between the policy and a reward-weighted distribution. While reverse KL is effective for refining a known good mode, it is inherently *mode-seeking*: it amplifies the first successful trajectory encountered rather than preserving a distribution over diverse high-reward solutions. In contrast, forward KL minimization exhibits *mode-covering* behavior, encouraging the policy to match the full support of the target distribution (Murphy, 2012). Although prior work has empirically observed declining diversity during training (Yue et al., 2025; Song et al., 2025; Chen et al., 2025a), the role of reverse KL as the root cause of mode collapse in RLVR has not been systematically addressed.

Motivated by this analysis, we propose *Distribution-*

[1]Tongji University, China [2]Independent [3]Shanghai AI Lab, China [4]Zhejiang University, China [5]Fudan University, China [6]The Chinese University of Hong Kong, China. Correspondence to: Qingwen Liu <qliu@tongji.edu.cn>.

*Proceedings of the 43$^{rd}$ International Conference on Machine Learning*, Seoul, South Korea. PMLR 306, 2026. Copyright 2026 by the author(s).

*Matching Policy Optimization* (DMPO), a principled on-policy algorithm that mitigates mode collapse by approximating forward KL minimization. Directly optimizing forward KL is intractable, as it requires sampling from a global reward-weighted distribution over trajectories. DMPO circumvents this challenge by operating at the *group level*: for each group of sampled trajectories, we construct a target distribution proportional to their rewards (a Boltzmann distribution) and explicitly align the policy distribution to this target. This formulation is naturally compatible with GRPO's group-based advantage normalization and can be implemented by adding a single distribution-matching regularization term to the standard objective. As a result, DMPO preserves multiple high-reward modes and sustains exploration throughout training.

To rigorously evaluate exploration behavior, we require tasks where multiple high-quality solutions exist, mode collapse is easily observable, and solution quality can be precisely measured. NP-hard combinatorial optimization satisfies all three criteria. Such problems admit exponentially many feasible solutions, but only a small fraction approach optimality, making them an ideal testbed for studying the trade-off between exploration and exploitation. Building on NP-Bench (Li et al., 2025), we introduce MM-NP-Bench, a multimodal benchmark comprising 10 text-based and visual combinatorial optimization tasks spanning *Routing*, *Covering*, *Partitioning*, *Substructure*, and *Constraints*. Each task includes parametric generators with controllable difficulty, exact rule-based verifiers, and heuristic solvers that provide near-optimal baselines, enabling precise evaluation of the solution quality via our proposed Quality Ratio metric.

Across both text-only and multimodal benchmarks, DMPO consistently outperforms GRPO and its variants. On NP-Bench, DMPO achieves a Quality Ratio of 43.9% compared to GRPO's 40.1%, a 9% relative improvement. On MM-NP-Bench, DMPO achieves 43.1% versus 38.4%, a 12% relative improvement. These gains indicate that distribution matching effectively prevents premature convergence, enabling discovery of higher-quality solutions that mode-seeking objectives fail to uncover. Moreover, the benefits extend beyond combinatorial optimization: DMPO improves mathematical reasoning by 2.0% and out-of-domain tasks by 2.3%, suggesting that diversity-preserving training yields more robust and transferable reasoning abilities.

In summary, our contributions are threefold:

- **Identifying and solving mode collapse:** We show that on-policy RL methods suffer from mode collapse due to reverse KL's mode-seeking behavior, and propose DMPO—a simple, practical solution that approximates forward KL minimization at the group level, achieving 9-12% relative improvements on optimization tasks.

- **MM-NP-Bench: A multimodal benchmark for testing

exploration:** We introduce MM-NP-Bench, extending NP-Bench (Li et al., 2025) to vision-language models with visual representations of 10 NP-hard tasks. The benchmark features dual-metric evaluation (Success Rate & Quality Ratio) that makes mode collapse observable: high SR but low QR reveals a policy that finds solutions but doesn't optimize them. We provide a complete infrastructure including parametric generators, rule-based verifiers, and heuristic solvers, enabling both evaluation and RLVR training.

- **Empirical validation and transfer learning:** Extensive experiments showing DMPO outperforms five strong baselines by 4.7%-3.8% on optimization tasks, 2% on mathematical reasoning, and 2.3% on out-of-domain tasks, with evidence that diversity-preserving training transfers to general reasoning capabilities.

## 2. Related Work

**Mode Collapse in Reinforcement Learning**   Mode collapse—the tendency of policies to concentrate on a single solution—is a well-documented phenomenon in reinforcement learning. The connection to reverse KL divergence's mode-seeking behavior is established in the literature: Murphy (2012) show that minimizing $D_{\mathrm{KL}}(q\|p)$ causes $q$ to concentrate on a single mode of $p$, while Levine (2018) explicitly discusses this in the context of maximum entropy RL. However, despite this theoretical understanding, mode collapse remains pervasive in practice. Ahmed et al. (2019) document decreasing entropy during policy optimization, noting that standard RL methods converge to near-deterministic policies. Eysenbach & Levine (2021) show that entropy regularization alone is insufficient to prevent collapse in multimodal reward landscapes. In language model training, Yue et al. (2025) and Song et al. (2025) observe that RLVR methods reduce solution diversity compared to the base model. While the theoretical connection between reverse KL and mode-seeking is known, *practical solutions remain elusive*. Forward KL minimization $D_{\mathrm{KL}}(p\|q)$ would provide mode-covering behavior, but requires sampling from the target distribution $p$, which is intractable.

**Diversity in Policy Optimization**   Several approaches have been proposed to maintain diversity in RL. **Entropy regularization** (Haarnoja et al., 2018; Ahmed et al., 2019) adds an entropy bonus to the reward, but Eysenbach & Levine (2021) show this is insufficient to prevent mode collapse when the policy becomes confident about a suboptimal solution. **Intrinsic motivation** (Pathak et al., 2017; Burda et al., 2018) augments rewards with exploration bonuses based on prediction error or state visitation counts, encouraging the policy to visit novel states, but these methods focus on state-space coverage rather than solution diversity and can be distracted by irrelevant novelty. **Ensemble meth-

**ods** (Elliott & Anderson, 2021; Zhumabekov et al., 2023) train multiple policies with different initializations, maintaining diversity across the ensemble, but scale poorly with ensemble size and lack theoretical guarantees for individual policy diversity. Recent work in LLM training has explored diversity from different angles. Cheng et al. (2025); Cui et al. (2025); Wang et al. (2025) constrain updates on high-covariance tokens to preserve stochasticity during training. Chen et al. (2025a) explicitly optimize pass@k metrics as rewards, encouraging the model to generate diverse responses where at least one succeeds. Zhuang et al. (2025); An et al. (2025) propose adaptive temperature strategies that vary sampling temperature across different prompts or training stages to balance exploration and exploitation. Zhu et al. (2025) propose FlowRL, which applies GFlowNet principles by learning a global partition function to construct a Boltzmann target distribution. However, FlowRL still minimizes reverse KL $D_{\mathrm{KL}}(\pi_\theta \| p^*)$ due to the intractability of sampling from the target distribution, and thus remains susceptible to mode-seeking behavior. Our work differs in that we approximate forward KL minimization $D_{\mathrm{KL}}(p^* \| \pi_\theta)$ at the group level—a tractable approximation that provides mode-covering behavior without requiring global sampling or a learned partition function. This approach seamlessly integrates with GRPO's group-based advantage normalization and demonstrates consistent improvement.

**RLVR for Language Model Reasoning** Reinforcement Learning with Verifiable Rewards (RLVR) has become the dominant paradigm for training reasoning models (Zhang et al., 2025; Guo et al., 2025; OpenAI, 2024). On-policy methods are particularly popular due to their stability and sample efficiency. PPO (Schulman et al., 2017) uses clipped importance ratios to constrain updates. GRPO (Shao et al., 2024b) extends PPO with group-based advantage normalization, improving stability for language models. GSPO (Zheng et al., 2025) further refines this with guided search. GPG (Chu et al., 2025) incorporates policy gradients with generalized advantage estimation. However, all these methods minimize reverse KL divergence, making them susceptible to mode collapse. Our work shows that a simple modification—adding a distribution-matching term—prevents this collapse while maintaining the stability and efficiency of on-policy training. DMPO seamlessly extends GRPO's group-based formulation, making it practical for large-scale language model training.

## 3. Distribution-Matching Policy Optimization

We propose DMPO, which mitigates mode collapse by approximating forward KL at the group level. We first show that standard on-policy RL minimize reverse KL, which causes mode-seeking and premature convergence, then derive DMPO as a practical solution providing mode-covering

behavior through group-level distribution matching.

### 3.1. Preliminaries: On-Policy RL and Mode Collapse

**Group Relative Policy Optimization (GRPO).** Consider a policy $\pi_\theta$ and reference policy $\pi_{\mathrm{ref}}$. Given query $x$, we sample $G$ trajectories $\mathcal{O} = \{o_1, \dots, o_G\}$ from the current policy $\pi_{\theta_{\mathrm{old}}}$. GRPO (Shao et al., 2024b) optimizes the policy using a clipped surrogate objective with group-normalized advantages. For trajectory $o_i$, the objective is:

$$\mathcal{J}_i(\theta) = \min\left(\rho_i(\theta)\hat{A}_i, \mathrm{clip}(\rho_i(\theta), 1-\epsilon, 1+\epsilon)\hat{A}_i\right), \quad (1)$$

where $\rho_i(\theta) = \pi_\theta(o_i|x)/\pi_{\theta_{\mathrm{old}}}(o_i|x)$ is the importance ratio, and $\hat{A}_i$ is the group-normalized advantage (z-score of rewards within the group): $\hat{A}_i = \frac{r(o_i) - \mathrm{mean}(\{r_j\})}{\mathrm{std}(\{r_j\}) + \epsilon}$. The complete GRPO objective is:

$$\mathcal{L}_{\mathrm{GRPO}}(\theta) = -\frac{1}{G}\sum_{i=1}^{G}\mathcal{J}_i(\theta) + \beta_{\mathrm{KL}}D_{\mathrm{KL}}(\pi_\theta\|\pi_{\mathrm{ref}}), \quad (2)$$

where $\beta_{\mathrm{KL}}$ is a hyperparameter controlling the strength of the KL penalty to the reference policy, preventing the policy from drifting too far from $\pi_{\mathrm{ref}}$ for training stability.

**Policy Gradient as Reverse KL Minimization.** To understand why mode collapse occurs, we establish the connection between policy gradient methods and reverse KL divergence. In Maximum Entropy RL (Ziebart et al., 2008; Levine, 2018), the objective is to maximize expected reward plus entropy: $\max_\theta \mathbb{E}_{\tau\sim\pi_\theta}[r(\tau) + \alpha H(\pi_\theta)]$, where $\alpha > 0$ is the temperature and $H(\pi_\theta) = -\mathbb{E}[\log\pi_\theta(\tau)]$ is the entropy. This is equivalent to minimizing reverse KL divergence to a Boltzmann target (Levine, 2018):

$$\min_\theta D_{\mathrm{KL}}(\pi_\theta\|p^*) \quad \text{where} \quad p^*(\tau) = \frac{\exp(r(\tau)/\alpha)}{Z}, \quad (3)$$

with partition function $Z = \sum_\tau \exp(r(\tau)/\alpha)$. Since $D_{\mathrm{KL}}(\pi_\theta\|p^*) = -H(\pi_\theta) - \mathbb{E}[r(\tau)/\alpha] + \log Z$ and $\log Z$ is constant in $\theta$, minimizing the KL is equivalent to maximizing $\mathbb{E}[r(\tau)] + \alpha H(\pi_\theta)$.

Standard policy gradient methods like GRPO do not include explicit entropy regularization, corresponding to $\alpha \to 0$. In this limit, the target becomes $p^*(\tau) \propto \mathbb{I}[\tau = \arg\max r(\tau)]$—a Dirac delta on the highest-reward trajectory. Thus, policy gradient methods implicitly minimize reverse KL, inheriting its mode-seeking behavior. While GRPO includes $D_{\mathrm{KL}}(\pi_\theta\|\pi_{\mathrm{ref}})$ to prevent drift from the reference policy, this does not address mode-seeking toward high-reward trajectories—the policy still collapses to a single mode within the high-reward region.

### 3.2. Forward KL Minimization: The Ideal Solution

Having established that reverse KL causes mode collapse through mode-seeking, the natural solution is forward KL

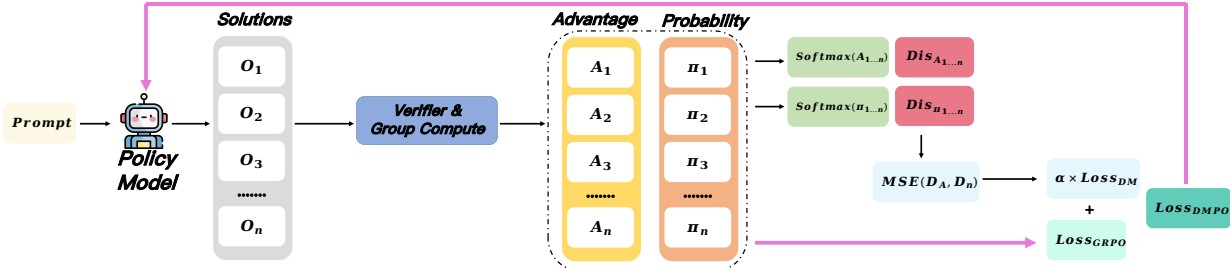

*Figure 1.* **Overview of DMPO.** The framework generates a set of trajectories $\{O_i\}$ to compute advantages $A$ and probabilities $\pi$. While standard **GRPO** (bottom flow) focuses on reward maximization, our novel **Distribution Matching** module explicitly aligns the policy's empirical distribution with the reward-induced target landscape via MSE. This dual-objective mechanism mitigates mode collapse, common in on-policy RL, and preserves the reasoning diversity crucial for solving complex reasoning problems.

divergence $D_{\text{KL}}(p^*\|\pi_\theta)$, which is mode-covering (Murphy, 2012). Forward KL penalizes placing low probability where the target has high probability, forcing the policy to cover all modes of $p^*$ and preventing mode collapse.

However, forward KL is intractable:

$$D_{\text{KL}}(p^*\|\pi_\theta) = \mathbb{E}_{o\sim p^*}\left[\log\frac{p^*(o)}{\pi_\theta(o)}\right]. \quad (4)$$

This requires sampling from $p^*$, which needs computing the partition function $Z = \sum_o \exp(r(o)/\alpha)$—exponential in trajectory length. For TSP with 20 cities ($\sim 10^{18}$ tours), this is infeasible. Standard on-policy RL therefore uses reverse KL, which only requires sampling from $\pi_\theta$.

### 3.3. Group-Level Approximation of Forward KL

Our key insight is to approximate forward KL at the *group level* rather than globally. For a group of trajectories sampled from $\pi_\theta$ for prompt $x$, we construct a group-conditional target distribution and match the policy distribution to it.

**Group-Level Target Distribution.** Given a group $\mathcal{O} = \{o_1, \ldots, o_G\}$, we define the group-conditional Boltzmann distribution:

$$p(o_i \mid \mathcal{O}) = \frac{\exp(r(o_i)/\alpha)}{\sum_{j=1}^{G}\exp(r(o_j)/\alpha)}. \quad (5)$$

This distribution represents the ideal probability allocation over the group—trajectories with higher rewards receive higher probability, but all trajectories maintain non-zero probability. Crucially, this only requires the *group-level partition function* $Z_{\text{group}} = \sum_{j=1}^{G}\exp(r(o_j)/\alpha)$, which is trivially computable.

**Group-Level Policy Distribution.** To measure the policy's actual probability allocation over the group, we must account for length bias. Since $\pi_\theta(o) = \prod_{t=1}^{|o|}\pi_\theta(o_t|o_{<t})$ decays exponentially with trajectory length, we use length-normalized log-likelihood: $\phi(o_i) = \frac{1}{|o_i|}\sum_{t=1}^{|o_i|}\log\pi_\theta(o_{i,t} \mid$

$o_{i,<t}, x)$. We then define the group-level policy distribution:

$$q_\theta(o_i \mid \mathcal{O}) = \frac{\exp(\phi(o_i))}{\sum_{j=1}^{G}\exp(\phi(o_j))}. \quad (6)$$

This represents the policy's relative preference over the group, independent of trajectory length.

**Distribution-Matching Loss.** We align the policy distribution $q_\theta$ with the target distribution $p$ by minimizing their divergence. The natural choice would be forward KL divergence $D_{\text{KL}}(p\|q_\theta)$, but we use mean squared error (MSE) for numerical stability:

$$\mathcal{L}_{\text{DM}}(\theta) = \frac{1}{G}\sum_{i=1}^{G}\left(p(o_i \mid \mathcal{O}) - q_\theta(o_i \mid \mathcal{O})\right)^2. \quad (7)$$

MSE provides bounded gradient coefficients. Since both $p$ and $q_\theta$ are probability distributions, $|p(o_i) - q_\theta(o_i)| \leq 1$ for all $i$. The gradient is:

$$\nabla_\theta\mathcal{L}_{\text{DM}} = \frac{2}{G}\sum_{i=1}^{G}\underbrace{(q_\theta(o_i) - p(o_i))}_{\text{bounded by }\pm 1}\nabla_\theta q_\theta(o_i), \quad (8)$$

ensuring stable training regardless of how the distributions evolve. In contrast, forward KL has gradient $-\sum_i \frac{p(o_i)}{q_\theta(o_i)}\nabla_\theta q_\theta(o_i)$, where the coefficient $p(o_i)/q_\theta(o_i)$ can grow arbitrarily large when $q_\theta(o_i) \to 0$. This is problematic when the policy becomes near-deterministic during training, potentially causing gradient explosion. Additionally, MSE approximates KL divergence via second-order Taylor expansion around $p = q$, providing similar optimization behavior when distributions are close while maintaining stability when they diverge.

**Unified Objective.** The complete DMPO objective combines reward maximization with distribution matching:

$$\mathcal{L}_{\text{DMPO}}(\theta) = \mathcal{L}_{\text{GRPO}}(\theta) + \lambda\mathcal{L}_{\text{DM}}(\theta). \quad (9)$$

Here, $\mathcal{L}_{\text{GRPO}}$ drives the policy toward higher rewards (exploitation), while $\mathcal{L}_{\text{DM}}$ maintains diversity by preventing collapse to a single mode (exploration). The hyperparameter $\lambda$ balances these two objectives.

### 3.4. Theoretical Analysis

We provide formal analysis of DMPO's properties, establishing guarantees on gradient stability, convergence behavior, and the exploration-exploitation tradeoff.

**Proposition 3.1** (Local Mode-Covering Behavior). *The distribution-matching loss $\mathcal{L}_{DM}$ provides mode-covering behavior within each group. Specifically, if trajectory $o_i$ has high reward ($r_i > 0$) such that $p(o_i \mid \mathcal{O}) > \epsilon$, then minimizing $\mathcal{L}_{DM}$ forces $q_\theta(o_i \mid \mathcal{O}) > 0$. More precisely, if $\mathcal{L}_{DM} < \delta$, then: $q_\theta(o_i \mid \mathcal{O}) \geq p(o_i \mid \mathcal{O}) - \sqrt{G\delta}$.*

*Proof.* By definition, $\mathcal{L}_{\text{DM}} = \frac{1}{G}\sum_i (p(o_i) - q_\theta(o_i))^2$. If $\mathcal{L}_{\text{DM}} < \delta$, then: $(p(o_i) - q_\theta(o_i))^2 < G\delta$. Taking square roots: $|p(o_i) - q_\theta(o_i)| < \sqrt{G\delta}$. Since $p(o_i) > \epsilon$, we have $q_\theta(o_i) > \epsilon - \sqrt{G\delta}$. For sufficiently small $\delta$, $q_\theta(o_i) > 0$. $\square$

Unlike reverse KL (which can place zero probability on some modes), DMPO's MSE loss explicitly prevents the policy from ignoring high-advantage trajectories in the group. This is the mathematical guarantee of mode-covering behavior.

**Proposition 3.2** (Convergence to Proportional Sampling). *In the limit $\lambda \rightarrow \infty$, if $\mathcal{L}_{DM} \rightarrow 0$, then: $\pi_\theta(o_i \mid x) \propto \exp(r_i/\alpha)$, meaning the policy samples trajectories proportionally to their exponentiated rewards.*

This shows DMPO achieves GFlowNet-like proportional sampling without requiring a backward policy.

**Proposition 3.3** (Smooth Interpolation via $\lambda$). *The hyperparameter $\lambda$ provides smooth interpolation between pure exploitation and diversity preservation:*

$$\lim_{\lambda \to 0} \mathcal{L}_{DMPO} = \mathcal{L}_{GRPO}, \quad \lim_{\lambda \to \infty} q_\theta \to p.$$

This allows practitioners to control the exploration-exploitation tradeoff via a single hyperparameter.

## 4. MM-NP-Bench: A Testbed for Diverse Reasoning

Having introduced DMPO as a solution to mitigate mode collapse, we now present MM-NP-Bench, a benchmark designed to validate DMPO's effectiveness. We need a testbed where mode collapse is easily observable, solution quality is precisely measurable, and diverse high-quality solutions exist. NP-hard combinatorial optimization provides these properties.

NP-hard problems feature **exponentially large multimodal solution spaces**—a TSP instance with 20 cities has $\sim 10^{18}$ valid tours with multiple local optima. A model that prematurely converges will settle for the first valid solution,

missing better alternatives. They provide **objective quality metrics**—precisely defined objectives (minimize tour length, maximize clique size) enable comparing solutions against heuristic baselines, transforming mode collapse into a quantifiable metric. They **decouple feasibility from optimality**—we can measure constraint satisfaction (Success Rate) and optimization quality (Quality Ratio) separately. Mode collapse produces a characteristic pattern: high SR (learns constraints) but low QR (fails to optimize).

NP-Bench (Li et al., 2025) provides text-based formulations of 10 NP-hard tasks. We extend it to the multimodal setting with visual representations, which offer three advantages: graph structures are more naturally expressed visually than through verbose adjacency lists, visual representations enable evaluation of vision-language models, and visual formats reduce specification ambiguity.

### 4.1. Benchmark Design

**Task Composition.**   MM-NP-Bench comprises 10 tasks across 5 categories: *Path Planning* (TSP, Hamiltonian Cycle), *Covering* (Vertex Cover, Dominating Set), *Graph Partition* (Minimum Cut, Maximum Cut), *Substructure* (Maximum Clique, Maximum Independent Set), and *Constraints* (Graph Coloring, Feedback Vertex Set). See Appendix C for complete task descriptions.

**Infrastructure.**   MM-NP-Bench provides complete infrastructure enabling both evaluation and RLVR training. The system consists of three integrated components working together to support the full research pipeline from instance generation to solution evaluation. Each task features: (1) parametric generators producing instances with textual, visual, and symbolic representations, (2) rule-based verifiers checking constraint satisfaction with $O(E)$ or $O(V^2)$ complexity, and (3) heuristic solvers (Table 6) computing near-optimal reference solutions. See Appendix C.2 for implementation details. Figure 1 illustrates how these three components work together: the generator creates instances with multiple representations, the verifier checks constraint satisfaction, and the heuristic solver computes reference solutions, enabling both rigorous evaluation and reward computation for RL training. A detailed task case from MM-NP-Bench is shown in Figure 3.

### 4.2. Evaluation Metrics

MM-NP-Bench uses dual-metric evaluation to make mode collapse quantitatively observable by decoupling constraint satisfaction from quality optimization.

**Success Rate (SR).**   The percentage of test instances where the model generates a valid solution satisfying all problem constraints. For example, in TSP, a valid solution must visit every city exactly once and form a closed

*Table 1.* Performance of reasoning MLLMs, general MLLMs, and our trained MLLMs on MM-NP-Bench.

| Model | Constraint | | Covering | | Partition | | Subgraph | | Path | | Overall | |
|---|---|---|---|---|---|---|---|---|---|---|---|---|
| | SR | QR | SR | QR | SR | QR | SR | QR | SR | QR | SR | QR |
| *Proprietary LLMs* | | | | | | | | | | | | |
| Gemini-3.0-Flash | 30.5 | 28.7 | **64.0** | **62.8** | **100.0** | 95.8 | 89.0 | 51.9 | **67.0** | **58.4** | **70.1** | **59.5** |
| o4-mini | 24.5 | 21.0 | 37.5 | 36.6 | 99.5 | 74.7 | **90.0** | 51.0 | 49.5 | 19.3 | 60.2 | 40.5 |
| GPT-4o-1120 | 12.5 | 8.4 | 4.0 | 3.3 | 47.5 | 34.5 | 66.0 | 28.5 | 19.5 | 7.1 | 29.9 | 16.4 |
| Qwen3-VL-235B-A22B-Thinking | 3.0 | 3.2 | 2.0 | 2.0 | 97.5 | 75.6 | 82.0 | 44.2 | 46.5 | 19.2 | 46.2 | 28.8 |
| Qwen3-VL-235B-A22B-Instruct | 5.0 | 4.7 | 6.0 | 4.9 | **100.0** | 76.1 | 71.0 | 36.1 | 39.5 | 18.4 | 44.3 | 28.0 |
| InternVL3.5-241B-A28B | 9.0 | 7.9 | 10.5 | 9.1 | 86.5 | 63.1 | 68.5 | 29.4 | 32.5 | 10.1 | 41.4 | 23.9 |
| *Open-Source LLMs* | | | | | | | | | | | | |
| Qwen3-VL-32B-Thinking | 1.5 | 1.1 | 13.5 | 12.0 | 95.5 | 69.1 | 76.0 | 39.6 | 36.5 | 14.4 | 44.6 | 27.2 |
| Qwen3-VL-32B-Instruct | 2.5 | 1.1 | 3.0 | 2.1 | 99.0 | 73.2 | 76.5 | 39.8 | 28.0 | 12.5 | 41.8 | 25.7 |
| Qwen3-VL-30B-A3B-Thinking | 8.5 | 6.6 | 2.5 | 1.7 | 95.0 | 69.3 | 69.5 | 33.9 | 34.0 | 13.9 | 41.9 | 25.1 |
| Qwen3-VL-30B-A3B-Instruct | 10.0 | 5.4 | 9.5 | 5.8 | 91.0 | 65.2 | 62.0 | 27.0 | 37.0 | 13.8 | 41.9 | 23.5 |
| Qwen3-VL-8B-Thinking | 0.5 | 0.5 | 0.0 | 0.0 | 71.5 | 50.6 | 59.5 | 22.6 | 32.5 | 11.2 | 32.8 | 17.0 |
| Qwen3-VL-8B-Instruct | 7.0 | 2.8 | 2.0 | 1.0 | 95.0 | 68.2 | 59.0 | 26.0 | 40.0 | 13.9 | 40.6 | 22.4 |
| InternVL3.5-38B | 3.5 | 2.9 | 5.0 | 3.7 | 98.0 | 70.1 | 51.5 | 18.9 | 24.5 | 7.3 | 36.5 | 20.6 |
| InternVL3.5-8B | 3.0 | 1.2 | 3.0 | 2.6 | 92.0 | 66.1 | 42.5 | 14.2 | 32.0 | 10.9 | 34.5 | 19.0 |
| Qwen2.5-VL-72B-Instruct | 11.0 | 4.8 | 6.5 | 3.4 | 95.5 | 67.9 | 56.0 | 18.7 | 32.0 | 10.8 | 40.2 | 21.1 |
| Qwen2.5-VL-32B-Instruct | 5.5 | 1.7 | 3.0 | 2.0 | 75.5 | 54.8 | 47.5 | 16.9 | 36.0 | 12.8 | 33.5 | 17.6 |
| Qwen2.5-VL-7B-Instruct | 12.5 | 4.6 | 19.5 | 10.5 | 66.0 | 38.6 | 50.0 | 15.0 | 4.5 | 1.6 | 30.5 | 14.1 |
| DMPO-VL-7B (Ours) | **62.5** | **31.5** | 25.5 | 22.1 | 98.5 | 73.1 | 72.0 | **67.4** | 51.0 | 21.6 | 61.9 | 43.1 |

tour. High SR indicates the model understands problem constraints and can generate feasible solutions, but says nothing about solution quality—a model could achieve high SR by always generating the same valid but suboptimal solution.

**Quality Ratio (QR).** For valid solutions, QR measures optimization quality relative to the heuristic solver's reference solution $V^*$:

$$QR(x_i, o_i) = \begin{cases} V_{\text{model}}(o_i)/V^*(x_i) & \text{(maximization tasks)} \\ V^*(x_i)/V_{\text{model}}(o_i) & \text{(minimization tasks)} \\ 0 & \text{(invalid solutions)} \end{cases}$$

where $V_{\text{model}}(o_i)$ is the objective value of the model's solution (e.g., tour length for TSP, number of colors for Graph Coloring).

QR directly measures a model's ability to discover high-quality solutions through exploration. A model that prematurely converges (mode collapse) will plateau at a suboptimal QR, while a model that maintains exploration will continue discovering better solutions, achieving higher QR. Success Rate complements this by ensuring methods don't sacrifice feasibility for quality—a method that improves both SR and QR demonstrates effective exploration that discovers more valid solutions and higher-quality ones.

### 4.3. Dataset and RLVR Training Setup

**Test Set.** 1,000 instances (100 per task) with difficulty calibrated to challenge state-of-the-art models. Difficulty is controlled via generation parameters (e.g., city count and spatial distribution for TSP, graph density for Graph Coloring).

**Training Sets.** We generate **NP-10K** (10,000 text-only instances) and **MM-NP-10K** (10,000 multimodal instances with visual representations). Training instances are filtered via rejection sampling to maintain SR $\in [0.05, 0.8]$ on base models, addressing the cold-start problem by ensuring sufficient positive rewards while preserving challenge.

**Reward Function.** For RLVR training, we define a composite reward encouraging both format compliance and solution quality: $R(o) = r_{\text{fmt}}(o) + r_{\text{opt}}(o)$, where $r_{\text{fmt}}(o) = 0.1$ if the output follows the required format (Chain-of-Thought reasoning followed by `Answer: [Solution]`), and $r_{\text{opt}}(o) = QR(o)$ if the solution is valid, 0 otherwise.

We use Quality Ratio (QR) rather than binary validity as the optimization reward for a crucial reason: binary rewards (1 if valid, 0 if invalid) offer no distinction between valid solutions of varying quality. In binary reward settings, all valid solutions are treated equally, providing no incentive for the model to improve quality once it finds a valid solution. This setup inevitably leads to mode collapse, as the model has no motivation to explore beyond the initial valid solution. In contrast, QR-based rewards offer a fine-grained signal: better solutions receive higher rewards, thereby encouraging further optimization. This reward structure is vital for studying mode collapse, as it ensures that the model receives the gradient signal needed for effective optimization. Addition-

*Table 2.* Performance comparison of various RLVR methods on MM-NP-Bench.

| Model | Constraint | | Covering | | Partition | | Subgraph | | Path | | Overall | |
|---|---|---|---|---|---|---|---|---|---|---|---|---|
| | SR | QR | SR | QR | SR | QR | SR | QR | SR | QR | SR | QR |
| Qwen2.5-7B-Instruct | 23.0 | 6.9 | 24.0 | 12.3 | 69.5 | 32.7 | 46.5 | 20.0 | 20.0 | 6.6 | 36.6 | 15.7 |
| GRPO (Shao et al., 2024b) | 54.5 | 21.2 | 18.0 | 17.7 | 87.0 | 67.4 | 68.0 | 65.9 | 51.0 | 19.6 | 55.7 | 38.4 |
| GSPO (Zheng et al., 2025) | 62.0 | 31.1 | 29.5 | **27.7** | 80.5 | 58.7 | 65.5 | 62.3 | 51.0 | 18.6 | 57.7 | 39.7 |
| GPG (Chu et al., 2025) | 52.0 | 23.7 | 22.5 | 17.4 | 78.4 | 62.4 | 71.4 | 64.1 | 48.5 | 16.6 | 54.6 | 36.8 |
| ClipCov (Cui et al., 2025) | 54.5 | 26.2 | 23.5 | 19.0 | 82.5 | 63.9 | 70.0 | 63.8 | **51.5** | 18.8 | 56.4 | 38.3 |
| FlowRL (Zhu et al., 2025) | 56.5 | 23.8 | 36.0 | 26.2 | 88.0 | 64.1 | 68.5 | 64.5 | 44.0 | 16.1 | 58.6 | 38.9 |
| GRPOPass@K (Meta) (Tang et al., 2025) | 60.5 | 20.7 | 42.5 | 21.1 | 78.5 | 56.0 | 56.0 | 53.7 | 42.5 | 14.9 | 56.0 | 33.3 |
| GRPOPass@K (Seed) (Chen et al., 2025b) | 64.5 | 23.4 | 47.0 | 18.4 | 83.0 | 56.4 | 73.5 | 59.6 | 45.5 | 16.1 | 62.7 | 34.8 |
| DMPO (Ours) | 62.5 | **31.5** | 25.5 | 22.1 | **98.5** | **73.1** | 72.0 | **67.4** | 51.0 | **21.6** | 61.9 | **43.1** |

*Table 3.* Performance comparison of various RLVR methods on six mathematical and text-only optimization reasoning benchmarks.

| Method | Math | | | | | | | NP-Bench | |
|---|---|---|---|---|---|---|---|---|---|
| | Math500 | OlympiadBench | Minerva | aime24 | aime25 | AMC | Average | SR | QR |
| Backbone | 63.6 | 28.3 | 21.7 | 14.0 | 6.9 | 45.6 | 30.00 | 29.6 | 14.6 |
| GRPO (Shao et al., 2024b) | 83.8 | 44.9 | 39.7 | 17.8 | 14.1 | 54.7 | 42.5 | 83.4 | 40.1 |
| GSPO (Zheng et al., 2025) | 83.2 | 44.7 | 39.3 | 18.8 | 12.6 | 53.9 | 42.1 | 82.6 | 41.2 |
| GPG (Chu et al., 2025) | 79.6 | 42.7 | 40.4 | 16.3 | 10.6 | 48.8 | 39.7 | **88.6** | 32.7 |
| ClipCov (Cui et al., 2025) | 81.4 | 46.7 | 38.6 | 19.1 | 13.8 | 54.5 | 42.3 | 87.2 | 41.3 |
| FlowRL (Zhu et al., 2025) | 82.4 | 44.0 | 37.6 | 16.0 | 11.1 | 50.2 | 40.2 | – | – |
| DMPO (Ours) | **85.6** | **48.4** | **41.2** | **20.2** | **15.0** | **56.3** | **44.5** | 85.3 | **43.9** |

ally, differences in achieved QR serve as a diagnostic tool, revealing whether methods successfully leverage the signal or converge prematurely. QR-based rewards also implicitly create a learning curriculum: models first learn to comply with the format (small, consistent rewards), then to satisfy constraints (positive QR), and ultimately to optimize quality (maximizing QR).

### 4.4. MM-NP-Bench Performance

To establish benchmark difficulty and demonstrate that MM-NP-Bench effectively differentiates model capabilities, we evaluate 17 state-of-the-art vision-language models on the test set without any RL training. Table 1 presents complete results across all tasks. Several key observations emerge from this baseline evaluation. First, even the best proprietary model (Gemini-3.0-Flash) achieves only 59.5% QR, indicating significant room for improvement and confirming the benchmark is challenging for frontier models. Second, the best open-source model (Qwen3-VL-235B-A22B-Thinking) achieves 28.8% QR, substantially below proprietary models, suggesting the benchmark effectively differentiates model capabilities. Third, all models exhibit substantial performance variation across task categories: Partition tasks (Graph Coloring, Bin Packing) show the highest QR (60-70%), indicating these are relatively easier to optimize, while Path tasks (TSP, VRP) and Constraint tasks (SAT, Graph Isomorphism) show the lowest QR (15-25%), indicating these require more sophisticated exploration strategies.

Fourth, model scaling shows diminishing returns: increasing model size from 7B to 72B parameters improves SR more than QR (relative improvement of 15% vs 8%), suggesting larger models learn constraint satisfaction more easily than optimization strategies. These baseline results validate MM-NP-Bench as a challenging testbed that reveals optimization limitations across diverse models and scales, providing an ideal environment for evaluating whether DMPO's distribution matching prevents mode collapse and enables discovery of higher-quality solutions.

## 5. Experiments

We validate DMPO through three research questions: (1) Does DMPO prevent mode collapse on optimization tasks? (2) Does DMPO generalize beyond optimization to mathematical reasoning? (3) Does diversity-preserving training transfer to out-of-domain tasks? We compare DMPO against five strong on-policy RL baselines: GRPO (Shao et al., 2024b), GSPO (Zheng et al., 2025), GPG (Chu et al., 2025), ClipCov (Cui et al., 2025), FlowRL (Zhu et al., 2025), GRPOPass@K (Meta) (Tang et al., 2025) and GRPOPass@K (Seed) (Chen et al., 2025b). Implementation details are provided in Appendix D.

**DMPO Prevents Mode Collapse.** Table 2 presents results on MM-NP-Bench. DMPO achieves **43.1% Quality Ratio versus GRPO's 38.4%**—a 12% relative improvement—while also achieving the highest Success Rate (61.9% vs. 55.7%). This improvement demonstrates that

*Table 4.* Performance on visual reasoning benchmarks. We compare models trained on: (1) baseline (no RL), (2) MM-NP-Bench only, (3) MM-16K only, and (4) their combination.

| Dataset & Method | LogicVista | GSM8K-V | MathVista | MathVision | MathVerse-V | Visulogic | WeMath | Average |
|---|---|---|---|---|---|---|---|---|
| Qwen2.5-VL-7B-Instruct (Backbone) | 44.5 | 13.6 | 68.3 | 24.5 | 37.3 | 22.6 | 33.1 | 34.8 |
| GRPO (MM-NP) | 44.5 | 13.2 | 71.3 | 25.1 | 39.0 | 25.3 | **39.5** | 36.8 |
| DMPO (MM-NP) | 46.1 | 15.1 | 69.3 | 26.9 | 40.7 | 25.9 | 35.4 | 37.1 |
| GRPO (MM-16K) | 45.2 | 18.3 | 71.7 | 26.6 | 32.5 | 25.8 | 37.2 | 36.8 |
| DMPO (MM-16K) | 48.1 | 13.7 | 72.3 | 26.5 | 43.3 | 25.4 | 34.8 | 37.7 |
| GRPO (MM-16K + MM-NP) | 45.6 | 15.5 | **72.8** | 27.0 | **47.8** | 24.1 | 42.4 | 39.3 |
| DMPO (MM-16K + MM-NP) | **48.7** | **18.5** | 72.3 | **28.0** | 46.0 | **26.0** | 39.0 | **39.8** |

DMPO's distribution matching prevents premature convergence, enabling discovery of higher-quality solutions that GRPO's mode-seeking behavior misses. DMPO's gains are consistent across task categories: +10.3% QR on Constraint tasks (21.2% → 31.5%), +5.7% on Partition tasks (67.4% → 73.1%), and +2.0% on Path tasks (19.6% → 21.6%). Notably, DMPO improves both metrics simultaneously—on Partition tasks, it achieves near-perfect SR (98.5%) while maintaining the highest QR (73.1%), demonstrating effective exploration discovers more valid solutions and higher-quality ones without trading off feasibility for quality. Comparing with FlowRL, another diversity-focused method, DMPO achieves substantially higher QR (43.1% vs. 38.9%) despite similar SR (61.9% vs. 58.6%), suggesting that DMPO's group-level Boltzmann target better maintains local diversity than FlowRL's learned partition function.

Table 3 shows DMPO achieves 43.9% QR on text-based NP-Bench versus GRPO's 40.1% (+9% relative improvement), demonstrating the approach is modality-agnostic. Notably, GPG achieves high SR (88.6%) but low QR (32.7%), indicating severe mode collapse where the model finds valid solutions but doesn't optimize them—DMPO's distribution matching prevents this pathology.

**Generalization to Mathematical Reasoning.** To demonstrate that DMPO's benefits extend beyond optimization, we evaluate on mathematical reasoning. We train Qwen2.5-Math-7B on Openr1-Math-46K and test on six benchmarks: AIME 2024, AIME 2025, AMC, Minerva Math, Olympiad-Bench, and MATH500. Table 3 (left) shows DMPO achieves 44.5% average accuracy versus GRPO's 42.5%—a 2.0% improvement. Gains are consistent across difficulty levels: +2.4% on AIME 2024 (hardest), +1.2% on AIME 2025, and +1.5% on OlympiadBench. These results suggest mathematical reasoning benefits from diversity preservation: many problems admit multiple valid proof strategies, and DMPO's mode-covering behavior enables exploration of different approaches rather than collapsing to the first successful method discovered.

**Transfer to Out-of-Domain Tasks.** To test whether diversity-preserving training on MM-NP-Bench enhances general reasoning capabilities, we evaluate on seven out-of-domain benchmarks spanning logic, visual reasoning, and mathematical problem-solving: LogicVista, GSM8K-V, MathVista, MathVision, MathVerse-V, VisuLogic, and WeMath. Table 4 shows training on MM-NP-Bench with DMPO improves out-of-domain performance by 2.3% on average versus the baseline, and outperforms GRPO by 0.3% (37.1% vs. 36.8%). This demonstrates that diversity-preserving training on optimization tasks transfers to general reasoning. Notably, when combined with general reasoning data (MM-16K), DMPO achieves 39.8% (+5.0% over baseline), suggesting optimization training and general reasoning are complementary.

**Ablation Studies.** We ablate $\lambda$ (distribution matching weight) and $\alpha$ (temperature) in Table 9 (Appendix D). Optimal values are $\lambda \in [0.3, 0.7]$ and $\alpha \in [0.8, 1.2]$, with performance slightly degrading outside these ranges. This shows DMPO is reasonably robust to hyperparameter choices.

**Different Divergence Settings.** As shown in Table 5, the DMPO loss can optimize the policy using only the distribution-matching term ($\lambda = \infty$), without the base GRPO loss, effectively removing the GRPO objective. This pure DMPO objective achieved a SR of 57.1% and a QR of 38.5%, slightly outperforming the standard GRPO (SR 55.6%, QR 38.4%). Additionally, we tested Jensen-Shannon (JS) divergence at the group level and found it effective in preventing mode collapse (SR 57.4%, QR 38.6%). However, both of these settings still underperform the full DMPO (MSE) objective. Therefore, DMPO should not be viewed merely as a local diversity regularizer. Rather, the strongest performance arises from combining the two objectives: GRPO provides a mode-seeking signal that drives reward maximization, while the DMPO term supplies a mode-covering signal that prevents collapse and preserves diversity across high-reward trajectories.

**Training Dynamics Analysis** Figure 5 (Appendix D.2) demonstrates that DMPO effectively mitigates the degenerative failure modes observed in GRPO. While GRPO suffers from both mode and length collapse—plateauing at suboptimal rewards and degenerating into trivial, short responses ($< 200$ tokens)—DMPO sustains continuous improvement and preserves robust reasoning chains ($\sim 400$ tokens). This structural integrity is further evidenced by

*Table 5.* Performance of DMPO under different variant settings on MM-NP-Bench.

| Method | Graph | | Schedule | | Partition | | Selection | | Planning | | Overall | |
|---|---|---|---|---|---|---|---|---|---|---|---|---|
| | SR | QR | SR | QR | SR | QR | SR | QR | SR | QR | SR | QR |
| Qwen2.5-7B-Instruct(backbone) | 11.0 | 3.1 | 40.0 | 19.8 | 67.0 | 34.0 | 26.7 | 15.2 | 3.5 | 1.0 | 29.6 | 14.6 |
| GRPO (Shao et al., 2024b) | 54.5 | 21.2 | 18.0 | 17.7 | 87.0 | 67.4 | **68.0** | **65.9** | 51.0 | 19.6 | 55.7 | 38.4 |
| DMPO w/o GRPO | 51.5 | 22.1 | 29.5 | 20.0 | 88.0 | 67.3 | **68.0** | 64.9 | 48.5 | 18.3 | 57.1 | 38.5 |
| DMPO w JS Divergence | 50.0 | 20.0 | 31.0 | 21.7 | 94.5 | **71.7** | 65.5 | 63.0 | 46.0 | 16.8 | 57.4 | 38.6 |
| DMPO w MSE Divergence | **85.3** | **25.6** | **82.0** | **60.3** | **100.0** | 52.8 | 67.7 | 55.0 | **81.5** | **25.6** | **83.3** | **43.9** |

the entropy dynamics: DMPO's lower entropy, paired with superior performance, indicates confident execution of logical steps (*structured diversity*) rather than the high-entropy stochastic confusion characteristic of GRPO.

# 6. Conclusion

In this work, we identify mode collapse as a key limitation of on-policy reinforcement learning in Large Reasoning Models, caused by reverse KL minimization's mode-seeking behavior, leading to premature convergence. We propose DMPO (Distribution-Matching Policy Optimization), which prevents mode collapse by approximating forward KL minimization at the group level. DMPO aligns the policy to a group-level target distribution, ensuring mode coverage without requiring global sampling. We introduce MM-NP-Bench, a multimodal benchmark extending NP-Bench with visual representations of 10 NP-hard tasks, designed to highlight mode collapse through dual-metric evaluation (Success Rate and Quality Ratio). Extensive experiments show DMPO's effectiveness: 4.7%-3.8% improvements on optimization tasks, 2.0% on mathematical reasoning, and 2.3% on out-of-domain tasks. These gains demonstrate that diversity-preserving training enhances both optimization and general reasoning capabilities. Our work establishes distribution matching as a practical solution to mode collapse in on-policy RL, with empirical validation showing that maintaining exploration leads to better solutions and more robust reasoning. Future work could combine DMPO with replay buffers or extend it to off-policy settings.

# Impact Statement

**Positive Impacts.** This work advances reinforcement learning methods for training Large Reasoning Models, with potential benefits including improved mathematical problem-solving, code generation, and optimization capabilities. By preventing mode collapse, DMPO enables more robust and diverse reasoning patterns, which could enhance AI systems' reliability in educational, scientific, and engineering applications. The MM-NP-Bench benchmark provides the research community with tools for evaluating and improving exploration in reasoning models.

**Potential Risks.** As with any advancement in large language model capabilities, improved reasoning abilities could be misused for generating misleading arguments, solving optimization problems for malicious purposes, or automating tasks in ways that displace human workers. However, these risks are not unique to our work and apply broadly to progress in AI reasoning capabilities.

**Broader Considerations.** We acknowledge that training large models requires substantial computational resources with associated environmental costs. Our method adds minimal overhead to existing training pipelines (single regularization term), limiting additional environmental impact. All experiments were conducted on shared research infrastructure to maximize resource efficiency.

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

# A. Method

## A.1. Algorithm

Algorithm 1 summarizes the complete DMPO training procedure.

---

**Algorithm 1** Distribution-Matching Policy Optimization (DMPO)

---

1: **Input:** Policy $\pi_\theta$, reference policy $\pi_{\text{ref}}$, dataset $\mathcal{D}$, group size $G$, hyperparameters $\lambda, \beta, \epsilon$
2: **for** each training iteration **do**
3:    Sample query $x \sim \mathcal{D}$
4:    Sample group $\mathcal{O} = \{o_1, \ldots, o_G\}$ from $\pi_{\theta_{\text{old}}}(o|x)$
5:    Compute rewards $\{r(o_i)\}$ and advantages $\{\hat{A}_i\}$ via z-score normalization
6:    Compute GRPO loss $\mathcal{L}_{\text{GRPO}}$ via Eq. 2
7:    Compute target distribution $p(o_i \mid \mathcal{O})$ via Eq. 5
8:    Compute policy distribution $q_\theta(o_i \mid \mathcal{O})$ via Eq. 6
9:    Compute distribution-matching loss $\mathcal{L}_{\text{DM}}$ via Eq. 7
10:    Update policy: $\theta \leftarrow \theta - \alpha \nabla_\theta(\mathcal{L}_{\text{GRPO}} + \lambda \mathcal{L}_{\text{DM}})$
11: **end for**
12: **Return:** Optimized policy $\pi_\theta$

---

# B. Limitations

First, DMPO currently relies on exact verifiable rewards, such as those in combinatorial optimization and mathematical reasoning. Extending it to open-ended tasks with subjective or learned rewards remains an open problem. Second, in extremely sparse-reward settings, if no valid solution appears in the sampled group, the local group approximation cannot recover the global optimum by itself. Replay or other off-policy extensions may help in this regime, and We will explore this direction in future work.

# C. MM-NP-Bench

## C.1. Overview

**Multimodal Solution Spaces.**   NP-hard problems feature exponentially large solution spaces with multiple high-quality solutions. For example, a TSP instance with 20 cities has approximately $10^{18}$ valid tours, many of which are feasible but vary dramatically in quality. Crucially, these solution spaces contain multiple local optima—regions where small modifications decrease quality but larger structural changes could lead to significantly better solutions. This creates natural pressure for sustained exploration: a model that prematurely converges to the first valid solution found will miss substantially better alternatives that require exploring through temporarily lower-reward regions.

**Objective Quality Metrics.**   Unlike open-ended reasoning where solution quality is subjective, optimization problems have precisely defined objectives such as minimizing tour length or maximizing clique size. This enables objective measurement of solution quality. We can compare a model's solution against reference baselines computed by heuristic solvers, quantifying exactly how close the model's solution is to optimal. This transforms mode collapse from a qualitative concern into a quantifiable metric that can be tracked during training and compared across methods.

**Decoupling Feasibility from Optimality.**   Crucially, optimization problems separate constraint satisfaction from quality optimization. We can measure two independent aspects: whether a solution is valid (satisfies all constraints) and whether it is high-quality (approaches optimality). A model exhibiting mode collapse demonstrates a characteristic pattern: it successfully learns to satisfy constraints but fails to optimize quality, settling for the first valid solution discovered. By measuring both aspects separately, we can directly observe and quantify this premature convergence.

NP-Bench (Li et al., 2025) provides a comprehensive text-based benchmark for combinatorial optimization, featuring 10 NP-hard tasks described in natural language. This establishes a strong foundation for evaluating optimization reasoning in language models. However, NP-Bench's text-only formulations create three limitations that motivate our multimodal extension. First, many optimization problems are inherently spatial and structural, making them more naturally expressed

visually than textually. A graph coloring problem is immediately interpretable from a visual rendering showing nodes and edges, whereas the text-only version requires parsing verbose adjacency lists. This verbosity increases cognitive load and conflates text parsing ability with optimization reasoning. Second, text-only benchmarks cannot evaluate vision-language models, which are increasingly deployed in real-world applications requiring multimodal reasoning. As these models advance, we need benchmarks assessing their ability to reason about visual optimization problems. Third, visual representations reduce specification ambiguity: edge weights displayed on a graph, node positions in a layout, and structural constraints in a diagram are unambiguous, whereas textual descriptions can be misinterpreted or require careful parsing.

### C.2. Bnechmark Details

MM-NP-Bench provides a robust infrastructure for evaluating MLLMs on visual combinatorial optimization tasks. It addresses the gap in optimization capability in existing reasoning benchmarks by focusing on NP-hard problems with complex solution landscapes. The task taxonomy is detailed below.

**Path Planning** These tasks assess the model's ability to perform sequential planning and understand global connectivity. We include: (1) `Traveling Salesman Problem (TSP)`: Finding the shortest Hamiltonian tour in a weighted graph. (2) `Hamiltonian Cycle`: Determining the existence of a path visiting every node exactly once. Success in this category requires long-horizon planning and spatial reasoning to avoid premature loops or dead ends.

**Covering Problems.** These tasks evaluate the identification of critical nodes that influence the entire graph structure. We include: (3) `Vertex Cover`: Selecting the minimum set of vertices such that every edge is incident to at least one selected vertex. (4) `Dominating Set`: Finding the minimum set of nodes such that every node in the graph is either in the set or adjacent to it. These problems test the ability to optimize global coverage through local node selection.

**Graph Partition.** These tasks require partitioning graphs based on relational density, testing the model's capacity to identify clusters and boundaries. We include: (5) `Minimum Cut`: Partitioning vertices to minimize the weight of crossing edges (identifying bottlenecks). (6) `Maximum Cut`: Partitioning vertices to maximize crossing edges (identifying bipartite structures). This category challenges the model to balance competing objectives between intra-cluster cohesion and inter-cluster separation.

**Substructure Discovery.** These tasks demand the extraction of specific patterns—either extremely dense or sparse—from complex visual noise. We include: (7) `Maximum Clique`: Finding the largest subset of vertices where every pair is connected (dense substructure). (8) `Maximum Independent Set`: Finding the largest subset of vertices with no connections between them (sparse substructure). These tasks act as duals, testing the model's ability to isolate signal from structural noise.

**Constraint Satisfaction.** Unlike pure optimization, these tasks involve strict validity rules where local violations render the solution invalid. We include: (9) `Graph Coloring`: Assigning labels to nodes such that no adjacent nodes share the same label, minimizing the total number of labels. (10) `Feedback Vertex Set`: Identifying the minimum set of vertices whose removal makes the graph acyclic (breaking all cycles). This category evaluates the model's ability to perform logical consistency checks and conflict resolution.

### C.3. Benchmark Creation Pipeline

**Parametric Generator.** For each task, we implement a controllable generation engine that produces instances based on difficulty parameters such as graph size, density, and constraint complexity. Each generated instance includes three synchronized representations. The textual specification provides a natural language description of the problem, objective, and constraints (e.g., "Find the shortest tour visiting all 20 cities exactly once"). The visual rendering depicts the problem structure as an image—for graph problems, this shows nodes, edges, and weights in a clear layout; for packing problems, this shows items and bins with size annotations. The symbolic representation provides structured data (adjacency matrices for graphs, coordinate lists for geometric problems, clause lists for SAT) enabling programmatic verification and deterministic evaluation. This multi-representation approach ensures models can be evaluated on their preferred input modality while maintaining consistent problem semantics across modalities.

**Rule-Based Verifier.** Each task has a deterministic verifier that checks constraint satisfaction with polynomial complexity ($O(E)$ or $O(V^2)$ depending on the task). Critically, verifiers check structural validity rather than accepting scalar outputs—for TSP, the model must output the complete tour sequence, not just the tour length. This prevents reward hacking

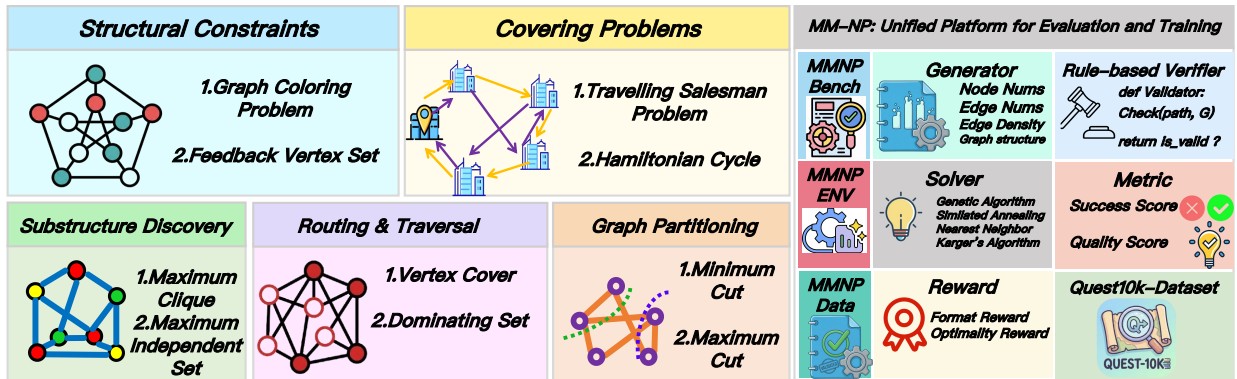

*Figure 2.* **Overview of MM-NP-Bench.** A multimodal benchmark for evaluating exploration in optimization reasoning. MM-NP-Bench features 10 NP-hard tasks with dual-metric evaluation (Success Rate and Quality Ratio) that makes mode collapse observable. Complete infrastructure includes parametric generators, rule-based verifiers, and heuristic solvers, enabling both evaluation and RLVR training.

where models might output plausible-looking numbers without actually solving the problem. Verifiers confirm all hard constraints are satisfied: for TSP, every city is visited exactly once and the tour forms a valid cycle; for Graph Coloring, no adjacent vertices share colors; for Bin Packing, no bin exceeds capacity. Only solutions passing verification are eligible for quality evaluation.

**Heuristic Solver.** For each generated instance, we run task-specific heuristic algorithms to compute a reference solution $V^*$ serving as the quality baseline. Table 6 lists the algorithms used for each task. These solvers are designed to be computationally efficient (polynomial or practical exponential time) while producing high-quality solutions. For example, we use 2-opt local search for TSP, which iteratively improves tours by swapping edges; Clarke-Wright savings algorithm for VRP, which constructs routes by merging profitable pairs; and greedy set selection for Set Cover, which iteratively selects sets covering the most uncovered elements. While these heuristics do not guarantee global optimality (NP-hard problems have no polynomial-time exact algorithms), they provide strong baselines that are difficult for models to match without sophisticated optimization reasoning. The reference value $V^*$ enables computation of Quality Ratio, quantifying how close a model's solution is to this near-optimal baseline.

*Table 6.* Heuristic solvers for the selected combinatorial optimization tasks. The solvers generally run in polynomial time or efficient exponential time for practical instances, providing baselines for the Quality Ratio.

| Task | Heuristic Algorithm | Complexity |
|---|---|---|
| *Constraint* | | |
| Graph Coloring (GCP) | DSATUR heuristic | $O(V^2)$ |
| Feedback Vertex Set | Greedy high-degree removal | $O(V \cdot E)$ |
| *Covering* | | |
| Vertex Cover | 2-approximation algorithm | $O(E)$ |
| Dominating Set | Greedy set cover reduction | $O(V^3)$ |
| *Partitioning* | | |
| Minimum Cut | Stoer-Wagner algorithm | $O(V \cdot E + V^2 \log V)$ |
| Maximum Cut | Greedy local search | $O(V \cdot E)$ |
| *Subgraph* | | |
| Maximum Clique | Bron-Kerbosch with pruning | Exponential (practical) |
| Max Independent Set | Min-degree greedy | $O(V \log V + E)$ |
| *Path* | | |
| TSP | 2-opt local search | $O(n^2)$ per iteration |
| Hamiltonian Cycle | DFS with pruning (or Pósa's) | Exponential (practical) |

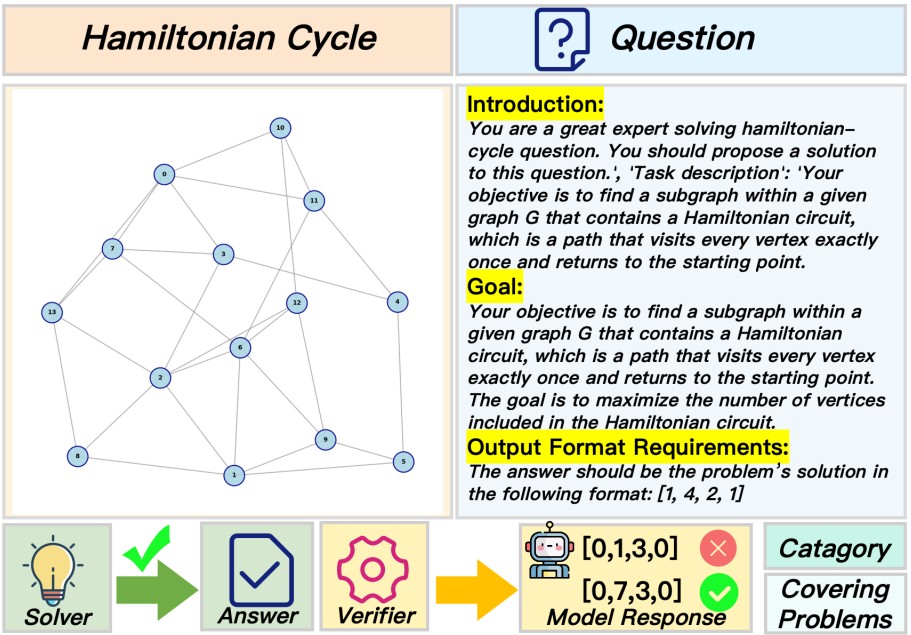

*Figure 3.* Example of a Hamiltonian Cycle in MM-NP-Bench. The goal is to find the longest cycle path in the graph under problem constraints. A rule-based verifier is used for precise evaluation, and a solver provides the optimal ground truth.

## D. Experiments

### D.1. Experimental Setup

**Model Architecture.** We use three base models depending on the task modality. For multimodal tasks (MM-NP-Bench and MM-16K (Meng et al., 2025)), we use Qwen2.5-VL-7B-Instruct. For mathematical reasoning, we use Qwen2.5-Math-7B trained on Openr1-Math-46K (Yan et al., 2025). For text-based optimization (NP-Bench), we use Qwen2.5-7B-Instruct-1M (Li et al., 2025).

**Training Data.** We train models on three datasets: (1) **MM-NP-Bench**: 10,000 multimodal optimization instances with visual representations, (2) **NP-Bench**: 10,000 text-only optimization instances, and (3) **Openr1-Math-46K**: mathematical reasoning data. For out-of-domain transfer experiments, we also use MM-16K (Meng et al., 2025), a general multimodal reasoning dataset.

**Evaluation Benchmarks.** We evaluate on three categories of benchmarks. **Optimization:** MM-NP-Bench (visual, 10 tasks) and NP-Bench (Li et al., 2025) (text, 10 tasks). **Mathematical reasoning:** Six benchmarks including AIME 2024, AIME 2025, AMC (Li et al., 2024), Minerva (Lewkowycz et al., 2022), OlympiadBench (He et al., 2024), and MATH500 (Hendrycks et al., 2021). We report avg@32 for AIME and AMC (smaller test sets), pass@1 for others. **Out-of-domain:** Seven visual reasoning benchmarks—LogicVista (Xiao et al., 2024), GSM8K-V (Yuan et al., 2025), MathVista (Lu et al., 2024), MathVision (Wang et al., 2024), MathVerse-V (Zhang et al., 2024), VisuLogic (Xu et al., 2025), and WeMath (Qiao et al., 2024)—evaluated using VLMEvalKit (Duan et al., 2025).

**Baseline Methods.** We compare DMPO against five strong on-policy RL baselines: GRPO (Shao et al., 2024b), GSPO (Zheng et al., 2025), GPG (Chu et al., 2025), ClipCov (Cui et al., 2025), FlowRL (Zhu et al., 2025), GRPOPass@K (Meta) (Tang et al., 2025), and GRPOPass@K (Seed). All methods use identical training data, model architecture, and hyperparameters (learning rate, batch size, training steps) following official VeRL configurations, with only method-specific parameters varying.

**Training Hyperparameters.** Common settings across all methods: group size $G = 8$, learning rate $1 \times 10^{-6}$ with cosine decay, 250 training iterations, clipping parameter $\epsilon = 0.2$, KL penalty $\beta_{\text{KL}} = 0.0$. DMPO-specific: distribution matching weight $\lambda = 2.0$, temperature $\alpha = \frac{1}{15}$. See Table 9 for complete settings and sensitivity analysis.

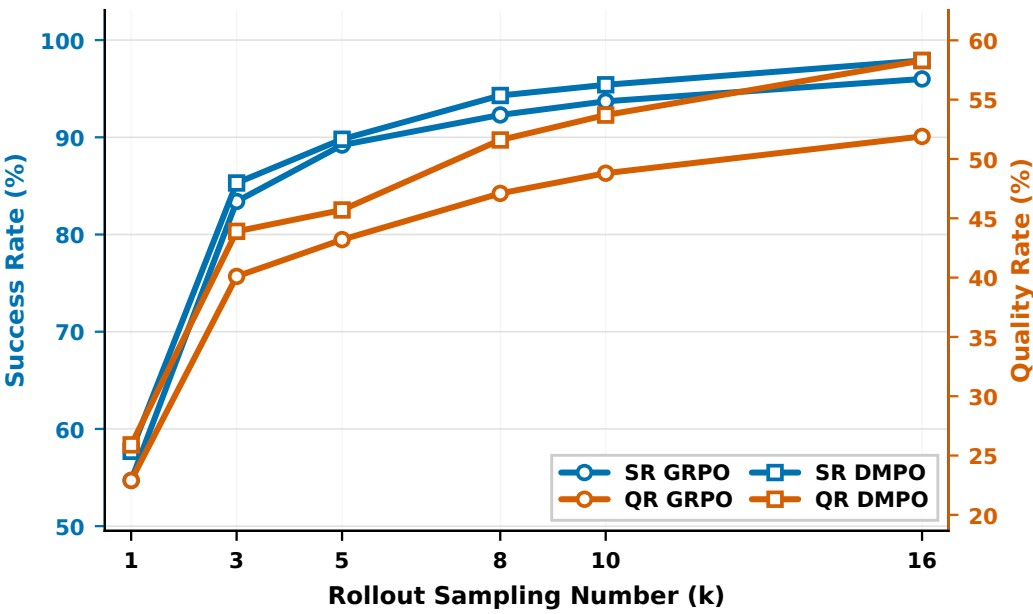

*Figure 4.* Performance of GRPO and DMPO on the MM-NP-Bench, showing Success Rate (SR) and Quality Ratio (QR) across different rollout sampling numbers ($k$).

*Table 7.* Computational comparison of different methods. All experiments use the same hardware, compute budget, and group size.

| Method | MFU | Time per Step (s) |
|---|---|---|
| GRPO (Shao et al., 2024b) | 0.0158 | 109.2 |
| GSPO (Zheng et al., 2025) | 0.0177 | 121.7 |
| GPG (Chu et al., 2025) | 0.0129 | 126.7 |
| ClipCov (Cui et al., 2025) | 0.0167 | 118.1 |
| FlowRL (Zhu et al., 2025) | 0.0670 | 67.3 |
| DMPO (Ours) | 0.0176 | 125.2 |

**Diversity Evaluation.** Our benchmark is designed to capture the distinction between feasibility and optimization quality. In this context, mode collapse manifests as a high Success Rate (SR) but a relatively low Quality Ratio (QR): the model consistently finds feasible solutions but repeatedly concentrates on a narrow set of suboptimal outcomes. To test whether DMPO simply collapses onto a single stronger optimum, we examine how SR and QR scale with the number of rollout samples ($k$). As shown in Figure 4, if a policy has collapsed to one dominant mode, increasing $k$ may still improve SR, but QR should plateau quickly because repeated samples yield essentially the same solution. This is precisely what we observe for GRPO: as $k$ increases from 1 to 16, SR rises sharply from 54.7% to 96.0%, while QR improves more gradually, saturating at 51.9%.

DMPO exhibits a markedly different pattern. As $k$ increases from 1 to 16, both SR and QR continue to improve steadily (SR: 57.7% to 97.9%, QR: 25.9% to 58.3%). This provides strong evidence that DMPO does not simply concentrate on a single dominant mode; rather, it maintains a broader set of high-quality solution modes, which can be effectively uncovered through additional sampling.

**Wall-Clock.** Although DMPO introduces the additional computation—the group-level distribution-matching MSE—compared to the base algorithm GRPO, it maintains a comparable computational efficiency. As shown in Table 7, the time per step and MFU of DMPO remain similar to those of GRPO and other baseline methods, indicating that the added MSE term does not introduce significant overhead while still improving the policy performance.

### D.2. Additional Experimental Results

**NP-Bench Analysis.** Results on NP-Bench (Table 8) further corroborate the effectiveness of DMPO in navigating complex combinatorial landscapes. A critical observation is the "optimality gap" prevalent in standard RLVR baselines; for instance,

while GPG achieves a high Overall Success Rate (88.6%), its Quality Ratio remains significantly lower (32.7%), indicating a tendency to settle for sub-optimal feasible solutions. This behavior is most pronounced in the *Schedule* task, where DMPO outperforms GPG by a substantial margin in QR (60.3% vs. 28.5%) despite a lower SR. This suggests that while greedy methods excel at satisfying hard constraints, they often fail to explore the high-reward regions of the solution space. In contrast, DMPO achieves the highest Overall Quality Ratio (43.9%), demonstrating its unique ability to transcend mere constraint satisfaction and engage in genuine global optimization. By aligning the policy with the reward-induced distribution, DMPO effectively mitigates the "winner-takes-all" pathology, ensuring that the model maintains the exploration pressure necessary to discover structurally superior solutions across diverse NP-hard domains.

*Table 8.* Performance of reasoning LLMs, general LLMs, and our trained LLMs on NP-Bench.

| Method | Graph | | Schedule | | Partition | | Selection | | Planning | | Overall | |
|---|---|---|---|---|---|---|---|---|---|---|---|---|
| | SR | QR | SR | QR | SR | QR | SR | QR | SR | QR | SR | QR |
| Qwen2.5-7B-Instruct(backbone) | 11.0 | 3.1 | 40.0 | 19.8 | 67.0 | 34.0 | 26.7 | 15.2 | 3.5 | 1.0 | 29.6 | 14.6 |
| GRPO (Shao et al., 2024b) | **93.7** | **30.5** | 69.0 | 49.6 | **100.0** | 51.8 | 63.3 | 44.3 | 91.0 | 24.3 | 83.4 | 40.1 |
| GSPO (Zheng et al., 2025) | 85.7 | 24.1 | 80.0 | 56.5 | 99.0 | 51.2 | 65.7 | 46.7 | 82.5 | **27.4** | 82.6 | 41.2 |
| GPG (Chu et al., 2025) | 92.7 | 23.5 | **96.0** | 28.5 | **100.0** | 51.8 | 59.3 | 39.6 | **95.0** | 20.3 | **88.6** | 32.7 |
| ClipCov (Cui et al., 2025) | 91.0 | 25.5 | 93.0 | 52.2 | 96.0 | 49.8 | **73.3** | 58.1 | 82.5 | 21.2 | 87.2 | 41.3 |
| DMPO (Ours) | 85.3 | 25.6 | 82.0 | **60.3** | **100.0** | **52.8** | 67.7 | 55.0 | 81.5 | 25.6 | 83.3 | **43.9** |

**Ablation Analysis.** To analyze the hyperparameter sensitivity of DMPO, we conduct an ablation study on the matching coefficient $\lambda$ and inverse temperature $\alpha$ (Table 9). The results reveal a fundamental trade-off between reward exploitation and distributional stability. The coefficient $\lambda$ controls the regularization strength; increasing $\lambda$ from 1.0 to 2.0 consistently improves both Success Rate (SR) and Quality Ratio (QR), indicating that a robust distributional anchor is essential to suppress the "winner-takes-all" pathology and prevent the model from settling for sub-optimal local modes. Meanwhile, the temperature $\alpha$ modulates the "sharpness" of the target landscape: a high $\alpha$ results in a nearly uniform target distribution that dilutes the optimization signal, while an excessively low $\alpha$ forces the target distribution more sharper, reintroducing mode collapse and negating the benefits of diverse exploration. Our optimal configuration of $\lambda = 2.0$ and $\alpha = \frac{1}{15}$ strikes a delicate balance: it provides sufficient gradient pressure to drive the policy toward high-quality solutions while maintaining a broad enough probability spread to sustain global search. This parameterization ensures that the policy frontier effectively mirrors the multi-modal reward landscape, leading to superior asymptotic performance across the complex tasks in MM-NP-BENCH.

**Training Dynamics Analysis** Figure 5 reveals the mechanism behind DMPO's improvements through training dynamics on MM-NP-Bench.

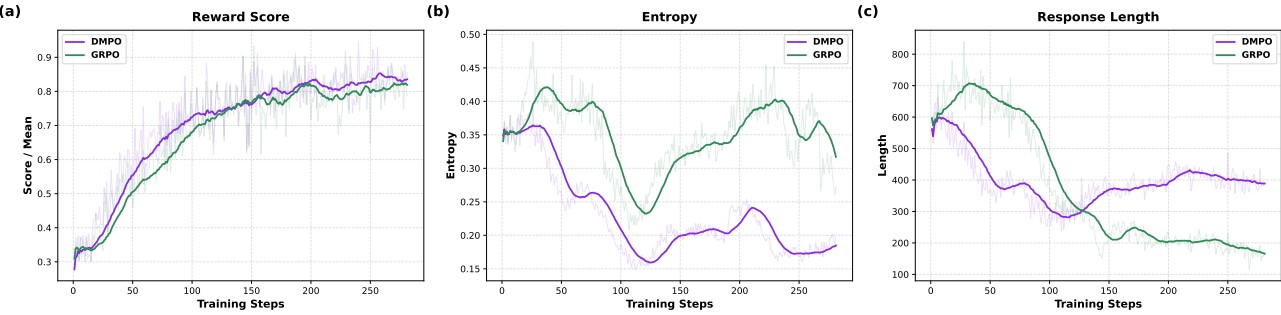

*Figure 5.* Comparison between GRPO and DMPO on training log dynamics. **Left (a)** shows the reward score, **Middle (b)** shows the entropy, and **Right (c)** shows the response length.

**Reward progression (Figure 5a):** DMPO achieves higher final rewards (0.85 vs. 0.75 for GRPO) and continues improving throughout training, while GRPO plateaus after approximately 100 steps. This plateau is the signature of mode collapse—the policy has concentrated on a suboptimal solution and stopped exploring despite the reward function incentivizing better solutions.

*Table 9.* Performance of DMPO with varying temperature $\beta$ and matching coefficient $\lambda$ on MM-NP-Bench.

| Parameter Settings | Constraint | | Covering | | Partition | | Subgraph | | Path | | Overall | |
|---|---|---|---|---|---|---|---|---|---|---|---|---|
| | SR | QR | SR | QR | SR | QR | SR | QR | SR | QR | SR | QR |
| $\lambda =1.0, \alpha = \frac{1}{15}$ | 58.5 | 28.2 | 23.5 | 19.0 | 88.5 | 68.9 | 70.0 | 65.2 | **51.5** | 18.8 | 58.4 | 40.0 |
| $\lambda =2.0, \alpha = \frac{1}{15}$ | 62.5 | 31.5 | 25.5 | 22.1 | **98.5** | **73.1** | **72.0** | 67.4 | 51.0 | **21.6** | 61.9 | 43.1 |
| $\lambda =2.0, \alpha = \frac{1}{5}$ | 53.5 | 27.6 | 24.5 | 21.8 | 89.0 | 68.3 | 80.5 | **77.0** | 43.0 | 15.3 | 58.1 | 42.0 |
| $\lambda =2.0, \alpha = \frac{1}{10}$ | **69.0** | **35.0** | **31.5** | **26.8** | 95.5 | 71.2 | 67.0 | 64.6 | 50.0 | 18.5 | **62.6** | **43.2** |
| $\lambda =2.0, \alpha = \frac{1}{20}$ | 55.0 | 29.3 | 22.0 | 18.8 | 94.5 | 68.5 | 59.5 | 58.4 | 47.0 | 16.0 | 55.6 | 38.2 |

**Length preservation (Figure 5c):** GRPO suffers from length collapse, where response length degenerates from $\sim$600 tokens to <200 tokens. This indicates the policy is taking shortcuts, generating minimal valid solutions rather than thorough reasoning chains. DMPO maintains $\sim$400 token responses throughout training, demonstrating that distribution matching acts as a regularizer preventing shortcuts.

**Entropy dynamics (Figure 5b):** Surprisingly, DMPO maintains lower entropy than GRPO despite being designed for exploration. In Chain-of-Thought reasoning, high entropy often indicates stochastic confusion (random token selection), while low entropy indicates confident execution of logical steps. DMPO's lower entropy combined with higher rewards and maintained length suggests it explores effectively through *structured diversity*—generating different reasoning strategies, not random tokens.

## D.3. Case Study of MM-NP-Bench

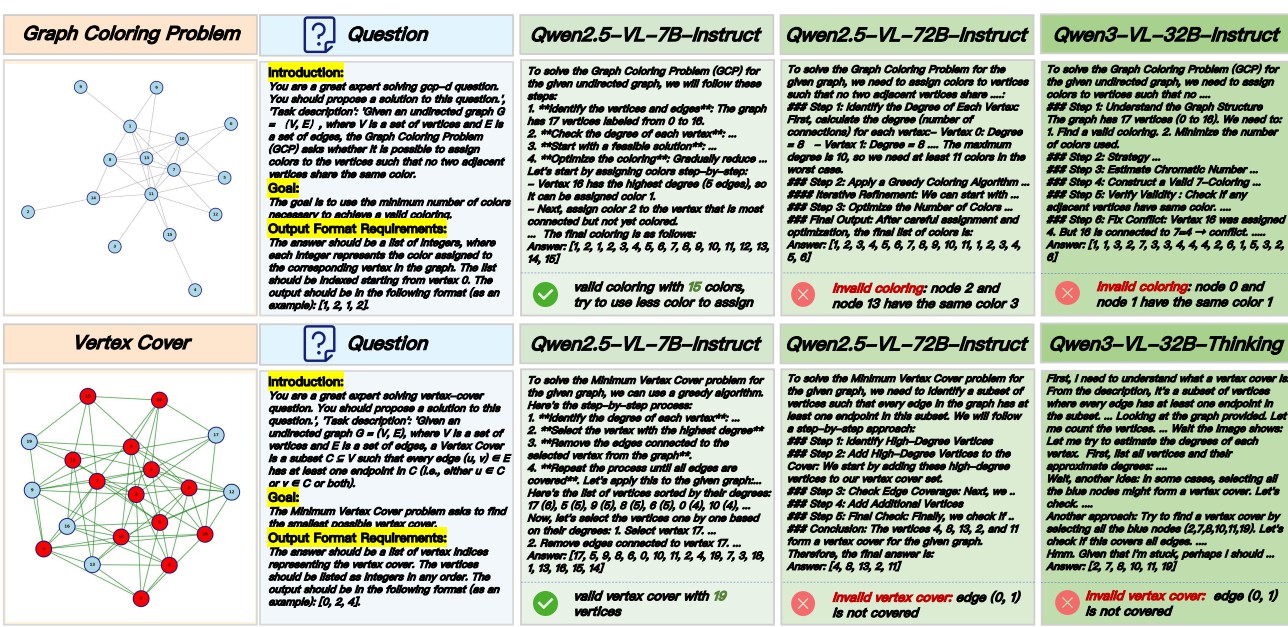

*Figure 6.* **Case Study of MM-NP-Bench.** Specifically, the Graph Coloring Problem task comes from the Constraint category, while Vertex Cover task comes from the Covering category.

Figure 6 presents a case study from MM-NP-Bench, where certain smaller models (e.g., Qwen2.5-VL-7B-Instruct) outperform larger or newer-generation models in specific tasks. We analyze this phenomenon through two sub-tasks: the Graph Coloring Problem and Vertex Cover.

In the Graph Coloring Problem, we observe that Qwen2.5-VL-7B-Instruct adopts a conservative exploration strategy. It initially attempts a solution using the maximum number of colors to ensure feasibility. When further optimization attempts fail, it intelligently reverts to the most reliable baseline solution. In contrast, while the other two models follow a step-by-step reasoning process, they tend to over-optimize a seemingly positive result while overlooking inherent flaws in the initial steps.

This premature optimization often triggers constraint violations (e.g., identical colors on adjacent vertices), leading to a lower success rate.

Furthermore, our analysis of the Vertex Cover task reveals critical insights into the Thinking models. While Qwen3-VL-32B-Thinking generates an extensive reasoning trace exceeding 4,000 tokens and significantly surpasses Qwen2.5-VL-7B-Instruct (500 tokens), this extended thought process appears to correlate with deeper hallucinations. A detailed inspection shows that the excessive reasoning length causes the model's visual perception to drift, leading to cumulative errors in reasoning. Ultimately, these layered mistakes result in an incorrect final output despite the extensive computation.

