# OpenReview forum: "Beyond Mode Collapse: Distribution Matching for Diverse Reasoning"
_ICML.cc/2026/Conference — ICML 2026 regular_

### Official Review · Reviewer_dQDn · 2026-03-10

**Soundness:** 3
**Presentation:** 3
**Significance:** 2
**Originality:** 2
**Overall Recommendation:** 4
**Confidence:** 4

**Summary:**

The manuscript titled "Beyond Mode Collapse: Distribution Matching for Diverse Reasoning",  studies why on policy RL methods for LLM/LRM reasoning (e.g., GRPO) tend to mode collapse concentrating probability mass on a single discovered solution and proposes Distribution Matching Policy Optimization (DMPO) to mitigate this failure. The core idea is to approximate forward KL minimization at the group level: for each prompt, construct a group conditional Boltzmann target over sampled trajectories proportional to rewards, and align the policy’s empirical group distribution to this target using an MSE distribution matching loss, combined with the usual GRPO objective. The approach aims to preserve mode covering behavior while remaining tractable and stable. Empirically, DMPO improves Quality Ratio (QR) and Success Rate (SR) on both a new multimodal NP hard benchmark (MM NP Bench) and the text NP Bench, and transfers to mathematical reasoning and out of domain visual reasoning tasks. Overall, this work explores a pertinent challenge.

**Compliance With Llm Reviewing Policy:**

Affirmed.

**Key Questions For Authors:**

Can you quantify how group size G and sampling temperature affect coverage of rare but high quality modes? Any empirical curves of QR vs. G
2Did you try forward KL with clipping or JS divergence at the group level? Any stability or performance trade offs vs. MSE?
If one used binary validity (SR only rewards) with DMPO, how much QR uplift would remain? (Teases apart the role of QR based reward vs. distribution matching.)
Plz report wall clock and FLOPs per method to validate fair comparisons, especially vs. FlowRL and GSPO.
are there any preliminary results combining DMPO with replay or off policy corrections?
It would help to report compute budgets and training durations per method to ensure improvements are not due to longer runs or implicit tuning advantages; some details exist but could be more explicit.

**Limitations:**

yes

**Strengths And Weaknesses:**

The paper connects reverse KL to mode seeking and shows why this amplifies the first high reward trajectory, giving an intuitive and formal rationale for collapse in GRPO style training
Simple, practical fix. DMPO’s group level Boltzmann target and MSE distribution matching integrate into GRPO with a single regularization term (λ L_DM), avoiding intractable global sampling while stabilizing gradients (bounded coefficients)
Strong empirical gains where exploration matters. On MM NP Bench, DMPO improves overall QR from 38.4% (GRPO) to  43.1% and SR from 55.7% to  61.9%, with category wise gains (e.g., +10.3% QR on Constraint tasks). On text NP Bench, QR improves 40.1% to 43.9%.
DMPO yields +2.0% average on math reasoning benchmarks and +2.3% average on out of domain visual reasoning, suggesting the learned diversity preserving behavior transfers. that shows the generalizations beyond optimization.
MM NP Bench offers a multimodal testbed with verifiers and heuristic solvers, and a dual metric design (SR, QR) that reveals mode collapse (high SR, low QR)
The paper analyzes λ and α sensitivity and shows training dynamics evidence (e.g., mitigating length collapse, maintaining longer reasoning chains), supporting the mechanism.
AS for as the weaknesses of the proposed method are concerned following may be addressed for improvements.
The method matches distributions only within sampled groups; while this is practical, it is an approximation to global forward KL and may not guarantee coverage outside the group’s support. Some discussion exists, but further analysis of bias relative to global p* would strengthen the theory.
The paper argues MSE ensures bounded gradients; however, empirical comparison to other divergences (e.g., forward KL with clipping, JS, χ²) is limited. A small study appears in theory text, but head to head ablations could bolster the choice.
While λ and α ranges are provided, defaults (e.g., λ=2.0, α≈1/15 in one place vs. α∈[0.8,1.2] sensitivity elsewhere) appear slightly inconsistent across sections; this could be unified for reproducibility.

---

> ### Author Rebuttal · Authors · 2026-03-31
>
> # Rebuttal to Reviewer dQDn
>
> We sincerely thank you for the "Weak Accept" recommendation and for recognizing the simplicity of DMPO, the strength of our empirical results, and the value of MM-NP-Bench. Your questions are highly constructive and help strengthen our empirical claims.
>
> **[1] Group Size ($G$), Rollout Temperature ($T$), and Coverage of Rare Modes**
>
> We agree that these two factors play distinct roles: the group size $G$ controls the resolution of DMPO's local distributional approximation, while the rollout temperature $T$ controls sampling diversity. We therefore ran ablation experiments over $G \in \{4, 8, 16\}$ and $T \in \{0.7, 1.0, 1.2\}$.
>
> As expected, increasing $G$ consistently improves both SR and QR, supporting our intuition that larger groups better approximate the target distribution and improve coverage of rare but high-quality modes:
>
> | Group Size ($G$) | SR   | QR   |
> |---|---:|---:|
> | 4  | 56.4 | 37.3 |
> | 8  | 61.9 | 43.1 |
> | 16 | 64.8 | 45.8 |
>
> The effect of rollout temperature is milder but still informative. Moderate exploration works best overall: $T=1.0$ gives the highest QR, while both lower and higher temperatures lead to slightly worse trade-offs.
>
> | Temperature ($T$) | SR   | QR   |
> |---|---:|---:|
> | 0.7 | 62.0 | 41.7 |
> | 1.0 | 61.9 | 43.1 |
> | 1.2 | 61.2 | 42.5 |
>
> Overall, DMPO benefits from larger group sizes while remaining fairly robust to moderate changes in rollout temperature.
>
> **[2] JS Divergence**
>
> This is an excellent suggestion. We initially chose MSE because its gradient coefficient is bounded in $[-1,1]$, which improves numerical stability. We also tested Jensen-Shannon (JS) divergence at the group level and found that it is effective as well in preventing mode collapse (SR: 57.4, QR: 38.6). We will include this ablation in the Appendix.
>
> **[3] Binary Validity Rewards (SR-only Ablation)**
>
> Using SR alone as a reward for NP-hard problems is inherently unstable, because trivially feasible but low-quality solutions are often easy to find. Without QR-based reward shaping, the policy lacks a meaningful signal for improving solution quality.
>
> That said, we can partially answer this question through our mathematical reasoning experiments, where the reward is strictly binary (1 for correct, 0 for incorrect). On these tasks, DMPO still yields a +2.0% average improvement over standard GRPO. This suggests that even under flat binary rewards among valid solutions, DMPO's distribution-matching term helps prevent collapse onto a single reasoning path and improves multi-sample metrics such as avg@32. We will clarify this point in the revision.
>
> **[4] Preliminary Results with Off-Policy Replay**
>
> You correctly identify that pure on-policy DMPO is bottlenecked if the global optimum is never sampled within the local group. To address this extreme sparse-reward regime, we explored combining DMPO with off-policy samples by mixing high-reward trajectories into the group-level target, which in principle can anchor the target distribution to known high-quality solutions.
>
> However, our preliminary investigations suggest that off-policy integration is not straightforward. In practice, replayed samples introduce substantial distribution shift, and simple importance weighting does not reliably correct this mismatch. This is consistent with recent findings in off-policy RL for LLMs [1], which show that stale data from past policies can destabilize optimization and sharply reduce policy entropy.
>
> Notably, although prior work studies off-policy data in the main RL objective, we observed a similar entropy collapse even when off-policy samples were introduced only through the FMR target. This drives the model toward over-exploitation and undermines the exploration benefits that DMPO is designed to provide.
>
> We therefore view off-policy replay as an important but technically demanding extension of DMPO. Stabilizing the FMR objective under off-policy shift will require additional algorithmic development and analysis. We will expand the discussion section accordingly and present off-policy DMPO as a key direction for future work.
>
> **[5] Compute Fairness, Wall-clock, and FLOPs**
>
> All comparisons are strictly fair: all methods use the same hardware, the same compute budget (281 steps), the same group size ($G=8$).
>
> | Method  | MFU    | Time per Step (s) |
> |---|---:|---:|
> | GRPO    | 0.0158 | 109.2 |
> | GSPO    | 0.0177 | 121.7 |
> | GPG     | 0.0129 | 126.7 |
> | ClipCov | 0.0167 | 118.1 |
> | FlowRL  | 0.0670 | 67.3 |
> | DMPO    | 0.0176 | 125.2 |
>
> **[6] Hyperparameter Consistency**
>
> We apologize for the typo regarding the optimal ranges of $\lambda$ and $\alpha$ in the main text. The correct values used in our main results are $\lambda=2.0$ and $\alpha=1/15$, fully consistent with Appendix C.
>
> # Reference
>
> [1] Xi, Zhiheng, et al. *BAPO: Stabilizing Off-Policy Reinforcement Learning for LLMs via Balanced Policy Optimization with Adaptive Clipping.* arXiv preprint arXiv:2510.18927 (2025).

---

> > ### Author Rebuttal · Reviewer_dQDn · 2026-04-03
> >
> > The rebuttal resolves most of my concerns, and I will maintain my current recommendation.

---

### Official Review · Reviewer_Rte7 · 2026-03-11

**Soundness:** 3
**Presentation:** 4
**Significance:** 4
**Originality:** 3
**Overall Recommendation:** 4
**Confidence:** 3

**Summary:**

The paper tries to tackle the well-known problem of mode-collapse in RL methods like GRPO, which affects the diversity of responses in finetuned LLMs. To do so, it proposes adding a forward KL divergence regularisation term to the objective function instead of the classical, tractable reverse KL term. To this end, the paper proposes DMPO (Distribution Matching Policy Optimization), which approximates the forward KL term at group level by a MSE term. They theoretically show the mode-covering behaviour induced by the MSE term. Then, they test DMPO on a proposed test-bed called MM-NP-Bench of NP-hard combinatorial problems in which the quality of solutions can be quantitatively measured. The extensive benchmarking on this dataset shows the increase in the quality of solutions when DMPO is employed.

**Compliance With Llm Reviewing Policy:**

Affirmed.

**Final Justification:**

Concerns about soundness were appropriately addressed during the rebuttal period, as a result of which I increased my soundness score to 3. The paper should now present a more theoretically sound approach to mode-seeking behaviour. It is a paper with clear merits, good originality and an excellent presentation. I am happy to maintain my weak accept score.

**Key Questions For Authors:**

1. Could you explain in more detail the relationship between the forward KL and the MSE objective used? They seem to be used interchangeably to describe different desired aspects of DMPO, but the link is not formally developed.
2. The paper \textit{Beyond Reverse KL: Generalizing Direct Preference Optimization with Diverse Divergence Constraints} by \textit{Wang et al.} or the paper \textit{Aligning Language Models with Preferences through f-divergence Minimization} by \textit{Go et al.} seem to discuss the concept of f-divergences, which already includes forward KL regularisations. How is your approach novel?

**Limitations:**

To my knowledge, limitations are not properly discussed in the paper. Concepts like generalisation of the method to other tasks is not mentioned.

**Strengths And Weaknesses:**

1. The paper is technically sound as a whole, with claims defended theoretically or empirically through an extensive benchmarking. The MM-NP-Bench is of special interest because it allows for a quantification of the quality of a reasoning model's solutions to problems. This allows for the claims about exploration of reasoning models to be made. However, some overly general claims are made:

1.a. Some claims have to be framed more precisely or notation needs to be consistent (or explain when you are purposely abusing notation). For example:

- ``Reverse KL causes mode collapse through mode-seeking, the natural solution is forward KL divergence which is mode-covering'' (lines 158-161, column 2) is too strong of a statement. Whilst this is true for specific cases, it is not true when the model class is sufficient rich. Indeed, papers such as \textit{KL-Regularized Reinforcement Learning is Designed to Mode Collapse} by \textit{Anthony GX-Chen} show that both reverse and forward KL regularisation can lead to policies which have multi-modal solutions. They explain how other factors such as the strength of the regularisation hyper-parameter control for the mode-seeking/mass-covering behaviour.

-  ``Since both $p$ and $q_{\theta}$ are probability distributions, $|p(o_i)-q_{\theta}(o_i)| \leq 1$" (lines 188-189) is only true for discrete probability distributions.

- ``MSE approximates KL divergence via second-order Taylor expansion around $p = q$'' (lines 201-202). Again misleading, the Taylor expansion has weighted coefficients, which is not the case of the MSE objective $\mathcal{L}_{\text{DM}}(\theta)$ presented.

- ``The gradient is [...] ensuring stable training regardless of how the distributions evolve" (lines 189-196). Only the gradient coefficients are bounded, why does this show that the training is stable?

2. The presentation is excellent. The sections follow a very natural order, with sequential and clear explanations which allows for an amenable reading. Some comments:

- From equation 7 to 8, probabilities go form $p(o_i| \mathcal{O})$ to $p(o_i)$. This is fine but you need to acknowledge it to not seem sloppy.

3. The paper addressed a critically important problem. RL-based methods for LLM finetuning are universal, so the mode-collapse problem which leads to a loss in the diversity of the model's responses is of central importance. The use of an approximate forward KL divergence term seems to be original and tackles the aforementioned undesirable behaviour.

---

> ### Author Rebuttal · Authors · 2026-03-30
>
> We sincerely thank Reviewer Rte7 for the "Weak Accept" recommendation and for recognizing the importance of addressing mode collapse, the value of MM-NP-Bench, and the clarity of our presentation. We especially appreciate your careful attention to the mathematical details. Your comments help us make our claims more precise, and we will revise the paper accordingly.
>
> **[1] Clarifying the Mechanics of Mode Collapse**
>
> Thank you for pointing us to the highly relevant work by Anthony GX-Chen et al. You are right that our original statement was too broad: reverse KL does not always imply mode seeking, nor does forward KL always imply mode covering. As GX-Chen et al. show, both can admit multimodal solutions depending on model flexibility, reward scale, and regularization strength.
>
> We will therefore soften our claim. Our intended point is that in the RLVR regime we study, especially with sparse or equal verifiable rewards and weak regularization, reverse-KL-style optimization has a strong inductive bias toward concentrating on the highest-support mode rather than covering all valid ones.
>
>
> This connection also clarifies DMPO. GX-Chen et al. identify a key failure case: when multiple correct answers receive the same reward, standard KL-regularized RL does not increase the relative probability of a lower-support correct mode. MARA addresses this by modifying the reward. DMPO addresses the same bottleneck from a distribution-matching perspective: we build a group-level Boltzmann target from observed rewards and match the policy to it, thereby redistributing probability mass toward all high-reward trajectories found in the rollout without reward shaping.
>
> **[2] Clarifying Technical Claims and Notation**
>
> You are correct on several technical points.
>
> First, the bound $|p(x)-q(x)| \le 1$ only holds in the discrete case. In our method, this condition applies because both $p(o_i|\mathcal{O})$ and $q_\theta(o_i|\mathcal{O})$ are categorical distributions over the finite sampled group $\mathcal{O}$. We will state this explicitly before Equation 8.
>
> Second, our wording around the Taylor expansion was imprecise. The exact local quadratic form of $D_{KL}(p\|q)$ around $p=q$ is a weighted squared error, not the unweighted MSE in Eq. 7. We deliberately drop the $\frac{1}{p_i}$ factor for stability, since it can explode when $p_i$ is very small. We will clarify that MSE is not the exact Taylor expansion, but a stable surrogate with the same optimum. Because the dropped factor is positive, it still preserves the per-coordinate update sign toward the target.
>
> Third, bounded coefficients do not imply globally stable LLM training. Only the scalar term $(p(o_i)-q_\theta(o_i))$ is bounded; the Jacobian $\nabla_\theta q_\theta(o_i)$ is not. Our intended claim was narrower: compared with exact forward KL, MSE avoids the specific coefficient explosion caused by $\frac{p(o_i)}{q_\theta(o_i)}$ when $q_\theta(o_i)$ is near zero.
>
> Finally, we agree that the notation change from Eq. 7 to Eq. 8 should have been stated explicitly. We will note that the conditioning on $\mathcal{O}$ is dropped there only for readability.
>
> **[3] The Relationship Between Forward KL and the MSE Objective**
>
> We agree that this link was under-explained. We will clarify that forward KL is our theoretical motivation, while MSE is our practical instantiation (providing stable online optimization).
>
> The connection is simple:
> (1) both objectives are non-negative and share the same global minimum at $p=q_\theta$;
> (2) the local quadratic form of forward KL is a weighted squared error, while our objective uses the unweighted version;
> (3) dropping the weight sacrifices exact local curvature but improves numerical stability while preserving the direction of the per-coordinate update.
>
> We will add this explanation in Section 3.3.
>
> **[4] Novelty Compared to f-Divergence DPO Methods**
>
> We appreciate this suggestion and will cite the relevant f-divergence DPO papers. Our setting is different: those methods operate in offline preference learning with a fixed dataset, whereas DMPO operates in online RLVR, where the model must sample trajectories, discover rewards, and construct a target distribution on the fly. Our method therefore addresses an online exploration problem rather than offline preference fitting.
>
> **[5] Limitations and Generalization**
>
> We agree that the limitations section should be expanded. In the revision, we will explicitly discuss two limitations.
>
> First, DMPO currently relies on exact verifiable rewards, such as those in combinatorial optimization and mathematical reasoning. Extending it to open-ended tasks with subjective or learned rewards remains an open problem.
>
> Second, in extremely sparse-reward settings, if no valid solution appears in the sampled group, the local group approximation cannot recover the global optimum by itself. Replay or other off-policy extensions may help in this regime, and We will explore this direction in future work.

---

> > ### Author Rebuttal · Reviewer_Rte7 · 2026-04-01
> >
> > Thank you for providing a detailed response to my review. I am pleased to see that all my concerns have been appropriately covered. Specifically:
> >
> > (i) Some theoretical claims have been soften to avoid over-promising.
> >
> > (ii) The small technical issues have been addressed appropriately. I also sincerely appreciate the extension of section 3.3 to explain more rigorously the relationship between forward KL and the MSE proxy used. Finally it is good to see that you plan on explicitly mentioning limitations of this method.
> >
> > (iii) Thank you for explaining the novelty compared to f-divergence methods.
> >
> > I am happy to maintain my weak accept score.

---

### Official Review · Reviewer_Fast · 2026-03-13

**Soundness:** 3
**Presentation:** 3
**Significance:** 3
**Originality:** 3
**Overall Recommendation:** 4
**Confidence:** 2

**Summary:**

This paper proposes Distribution-Matching Policy Optimization (DMPO) to address the mode collapse problem in GRPO for LLM reasoning tasks, especially NP-hard combinatorial optimization problems. A forward KL minimization loss is suggested to encourage mode coverage, and a group-level approximation is introduced to overcome the intractable global partition function. the distribution-matching loss (MSE for approximated target Q) is added to GRPO loss. In the experiments, DMPO demonstrates higher success rates and qauality ratios.

**Compliance With Llm Reviewing Policy:**

Affirmed.

**Final Justification:**

As noted in my comments, my questions are mostly resolved by the rebuttal. I encourage the authors to reflect in the manuscript the limitations regarding sparse-reward settings and off-policy extension. I will keep my current score.

**Key Questions For Authors:**

- In extreme sparse-reward settings, if the initial policy never samples the global optimum in its local group, how does DMPO avoid simply collapsing onto the best sub-optimal sample?

- Can the DMPO loss optimize the policy without the base GRPO loss? If not, is the proposed method fundamentally acting as a local diversity regularizer rather than true distribution matching?

**Limitations:**

No. While the paper presents a strong contribution, the discussion on limitations could be further enriched by acknowledging the inherent risks of combining on-policy exploration with local approximation in sparse-reward environments

**Strengths And Weaknesses:**

## Strengths
- **Solid motivation:** The paper tackles a very real and practical problem. Mode-seeking behavior in GRPO causes premature convergence in verifiable reasoning tasks, and I think it's a critical bottleneck in the field.
- **I like the idea of group-level approximation:** It's a practical way to enable forward KL minimization without the massive overhead of calculating a global partition function.

## Weaknesses
- **diversity evaluation:** The method still relies on the mode-seeking GRPO loss. Because of this, proving actual diversity is crucial. However, the chosen metrics (SR and QR) strictly measure the *quality* of the final discovered mode, leaving the core claim of "diverse reasoning" unsupported by trajectory-level metrics. For me, it's hard to tell if the model is actually maintaining a diverse distribution or just collapsing onto a better single optimum.
- **GFlowNet baseline:** The core motivation of DMPO, achieving forward KL distribution matching, is shared with Generative Flow Networks (GFlowNets) are designed to solve. Specifically, objectives like VarGrad [1] mathematically cancel out the partition function [2]; thus, it doesn’t compute an intractable global partition function. Even though they often require training a backward policy, they allow off-policy exploration (e.g., using replay buffers) to handle sparse rewards. An empirical comparison against these established distribution-matching frameworks would greatly strengthen the paper by providing a rigorous baseline for the proposed on-policy approach

--------
[1] Richter, Lorenz, et al. "Vargrad: a low-variance gradient estimator for variational inference." *Advances in Neural Information Processing Systems* 33 (2020)

[2] Zhang, David W., et al. "Robust scheduling with gflownets." ICLR 2023

---

> ### Author Rebuttal · Authors · 2026-03-31
>
> We sincerely thank you for the "Weak Accept" recommendation and for recognizing the practical importance of addressing mode-seeking behavior in GRPO. We also appreciate your thoughtful questions on diversity metrics and sparse-reward settings.
>
> **[1] Diversity Evaluation and Trajectory-Level Metrics**
>
> We agree that trajectory-level diversity can be informative, but in combinatorial optimization it can also be misleading: superficially different reasoning traces often map to the same final solution, whereas meaningful diversity lies in covering distinct high-quality solutions.
>
> Our benchmark is designed to capture this distinction by separating feasibility from optimization quality. Under this view, mode collapse appears as high SR but relatively low QR: the model reliably finds feasible answers, yet repeatedly concentrates on a narrow set of sub-optimal solutions. To test whether DMPO is simply collapsing onto a better single optimum, we examine how SR and QR scale with the number of samples ($k$). If a policy has collapsed to one dominant mode, increasing $k$ may still improve SR, but QR should plateau quickly because repeated samples recover essentially the same solution. This is what we observe for GRPO: as $k$ increases from 1 to 16, SR rises sharply from 54.7 to 96.0, while QR improves more slowly and begins to saturate, from 22.9 to 51.9.
>
> DMPO shows a different pattern. As $k$ increases from 1 to 16, both SR and QR continue to improve consistently (SR: 57.7 to 97.9, QR: 25.9 to 58.3). We view this as strong evidence that DMPO is not simply concentrating on a single stronger mode, but instead maintains a broader set of high-quality solution modes that can be uncovered through additional sampling.
>
> | Method | 1 | 3 | 5 | 8 | 10 | 16 |
> |---|---|---|---|---|---|---|
> | GRPO | 54.7; 22.9 | 83.4; 40.1 | 89.2; 43.2 | 92.3; 47.1 | 93.7; 48.8 | 96.0; 51.9 |
> | DMPO | 57.7; 25.9 | 85.3; 43.9 | 89.8; 45.7 | 94.3; 51.6 | 95.4; 53.7 | 97.9; 58.3 |
>
> **[2] GFlowNet Baseline Comparison**
>
> We agree that GFlowNets are a highly relevant framework for forward-KL-style distribution matching. In fact, we already include a strong GFlowNet-inspired baseline in our experiments: FlowRL, which constructs a Boltzmann target by learning a global partition function.
>
> DMPO achieves a substantially higher performance than FlowRL on both NP and Math reasoning benchmark (Table 2 and 3). This suggests that DMPO's group-level Boltzmann target better preserves local diversity and optimization pressure than FlowRL's learned global partition function.
>
> **[3] Extreme Sparse-Reward Settings and Local Optima**
>
> We thank the reviewer for raising this important challenge. If the global optimum is not sampled in the local group, DMPO still maintains a proportional distribution over the valid sub-optimal trajectories within that group, rather than collapsing all probability mass onto a single path. This keeps exploration open and reduces premature concentration on the "best of the worst." In our current setup, we also mitigate extreme cold-start failure by filtering training instances so that the base model has an initial SR in the range `[0.05, 0.8]`.
>
> That said, we agree that off-policy replay is a compelling direction for extreme sparse-reward regimes. We conducted preliminary explorations by mixing high-reward off-policy samples into the group target, but found this extension unstable in practice. Recent literature reports a similar issue in off-policy RL for LLMs [1]: using stale data from past policies can destabilize optimization and sharply reduce policy entropy.
>
> Notably, although prior work studies off-policy data in the main RL objective, we observed similar entropy collapse even when off-policy samples were introduced only through the DM term. This suggests that off-policy integration is not a trivial add-on. Stabilizing the matching distribution under off-policy shift requires additional algorithmic development and analysis, and we therefore view it as an important direction for future work.
>
> **[4] DMPO Loss Without Base GRPO**
>
> Yes, the DMPO loss can optimize the policy without the base GRPO loss. We ran an ablation using only the distribution-matching term ($\lambda = \infty$), which removes the GRPO objective. This pure DMPO objective achieved SR 57.1 and QR 38.5, slightly outperforming standard GRPO (SR 55.6, QR 38.4).
>
> However, it still underperforms the full DMPO objective. We therefore do not view DMPO as merely a local diversity regularizer. Rather, the strongest results come from combining the two objectives: GRPO provides a mode-seeking signal that drives reward maximization, while the DMPO term provides a mode-covering signal that prevents collapse and preserves diversity across high-reward trajectories.
>
> # Reference
>
> [1] Xi, Zhiheng, et al. "Bapo: Stabilizing off-policy reinforcement learning for llms via balanced policy optimization with adaptive clipping." arXiv preprint arXiv:2510.18927 (2025).

---

> > ### Author Rebuttal · Reviewer_Fast · 2026-04-03
> >
> > I appreciate the authors’ effort in addressing the concerns. My questions are mostly resolved by the rebuttal. I encourage the authors to incorporate into the manuscript the limitations regarding sparse-reward settings and off-policy extension discussed in the rebuttal. I will keep my score at weak accept.

---

### Official Review · Reviewer_NRHx · 2026-03-19

**Soundness:** 2
**Presentation:** 3
**Significance:** 3
**Originality:** 3
**Overall Recommendation:** 4
**Confidence:** 5

**Summary:**

This paper investigates the mode-seeking behavior of on-policy RL methods in the context of reasoning models. The authors first show that this behavior arises from the reverse KL minimization formulation commonly used in such methods.

Motivated by this observation, they explore an alternative perspective based on forward KL minimization, which is inherently mode-covering. To address the challenge of computing the global partition function, the authors propose approximating the target distribution using a group of sampled rollouts.

Building on this idea, they introduce Distribution Matching Policy Optimization (DMPO), which incorporates a distribution-matching objective into standard group-based RL (e.g., GRPO). Empirical evaluations on combinatorial optimization tasks (both unimodal and multimodal) as well as mathematical reasoning tasks demonstrate the effectiveness of the proposed approach.

**Compliance With Llm Reviewing Policy:**

Affirmed.

**Final Justification:**

The authors have addressed my key questions with additional experiments (please see strengths and weaknesses). Therefore, I am increasing my score by 1.

**Key Questions For Authors:**

[1] Comparison with pass@k optimization methods

The pass@k optimization method (Chen et al., 2025) is discussed in the related work as a diversity-focused approach, yet it is not included as an experimental baseline.

Given its relevance to diversity, it would be valuable to include this method in the comparisons.

[2] Hyperparameter sensitivity and consistency

There appears to be a mismatch in the discussion of sensitivity to $(\lambda, \alpha)$ between:
- The main paper (Ablation Studies, p.8), and
- The appendix (Training Hyperparameters, p.15), where Table 7 does not cover the broader range mentioned earlier.

What values of $\lambda$ and $\alpha$ are used for the main results?

How should these parameters be selected or tuned across different domains?

[3] Role of reverse KL in DMPO objective

If the forward KL objective is intended to encourage mode coverage, why is it necessary to combine it with the reverse KL (GRPO) objective in Eq. (9)?

Would optimizing only the distribution-matching term $L_\text{DM}$ (e.g., $\lambda \to \infty$) with an appropriate choice of $\alpha$ yield better or comparable results?

It would be helpful to include this as an ablation variant to better understand the role of the GRPO term in DMPO.

[4] Evaluation metrics for reasoning tasks

In Table 3, for mathematical reasoning tasks:
- Is performance reported using pass@1?
- How does the method perform under pass@k metrics (for larger k)?

**Limitations:**

yes

**Strengths And Weaknesses:**

**Strengths**

Clarity and presentation: The paper is well-written and easy to follow.

Important problem: The study of mode-seeking behavior is highly relevant, as it directly impacts the training dynamics and performance of reasoning models.

Novel and well-motivated approach: The connection between reverse KL (mode-seeking) and forward KL (mode-covering) is clearly articulated, and the proposed method is conceptually sound.

Practicality: The method is simple to implement, requiring only an additional regularization term on top of standard GRPO.

Empirical validation: The paper demonstrates consistent improvements across multiple domains.

**Weaknesses**

Missing relevant baselines: In particular, pass@k optimization methods, which are closely related to diversity, are not included in the empirical comparison.

Hyperparameter sensitivity: The performance appears to depend on key hyperparameters $(\lambda, \alpha)$, but guidance on how to choose them across domains is limited or unclear.

---

> ### Author Rebuttal · Authors · 2026-03-30
>
> # Rebuttal to Reviewer NRHx
>
> We sincerely thank Reviewer NRHx for recognizing the clarity of our presentation, the importance of addressing mode collapse, and the novelty and practical value of our DMPO approach. We also appreciate your careful reading and the opportunity to clarify our baselines, hyperparameter settings, and evaluation metrics.
>
> **[1] Comparison with pass@k optimization methods**
>
> We agree that pass@k optimization is a highly relevant direction for comparison. To address this point, we include two pass@k-style baselines: the method proposed in [1], and the pass@k objective officially supported in VeRL, which is derived from [2].
>
> In our experiments, the pass@k baseline underperformed DMPO. This result supports our central hypothesis: simply rewarding success within a batch is not sufficient to effectively prevent mode collapse. In contrast, DMPO provides an explicit mode-covering training signal by aligning the policy with a group-level Boltzmann target, which leads to substantially better diversity-quality trade-offs.
>
> | Method                | SQ    | QR    |
> |-----------------------|-------|-------|
> | GRPO_passk_seed [1]   | **62.7** | 34.8  |
> | GRPO_passk [2]        | 56.0  | 33.3  |
> | GRPO                  | 55.7  | 38.4  |
> | DMPO                  | 61.9  | **43.1** |
>
> These results suggest that while pass@k-style training can improve success quality in some cases, it does not consistently preserve reward diversity. DMPO, by contrast, achieves a much stronger balance between the two.
>
> **[2] Hyperparameter sensitivity and consistency**
>
> Thank you for catching this discrepancy. You are absolutely correct: the statement in the main text (Page 8) regarding the optimal hyperparameter range was inaccurate and remained from an earlier draft.
>
> For all main experiments, we used $\lambda = 2.0$ and $\alpha = 1/15$, following the FlowRL baseline setting in [3]. As reported in Appendix C (Table 7), we ablated $\lambda \in \{1.0, 2.0\}$ and $\alpha \in \{1/5, 1/10, 1/15, 1/20\}$. Across these settings, $\lambda = 2.0$ and $\alpha = 1/15$ yielded the best overall performance.
>
> **Practical guidance.** In practice, we recommend setting $\lambda = 2.0$ to keep the distribution-matching loss and the GRPO loss on a comparable scale. For the temperature, $\alpha \in \{1/5, 1/10, 1/15\}$ all produce consistently strong results, with $\alpha = 1/15$ being the best in our experiments. We will revise the text on Page 8 to exactly match Table 7 and will also include this tuning guidance in the main paper.
>
> **[3] Role of reverse KL in the DMPO objective**
>
> This is an excellent question. We ran exactly the ablation you suggested by optimizing only the distribution-matching term, i.e., setting $\lambda = \infty$ so that the GRPO objective is removed.
>
> We found that the pure distribution-matching objective (SR 57.1, QR 38.5 on MM-NP-Bench) slightly outperformed standard GRPO (SR 55.6, QR 38.4), but still underperformed the full DMPO objective (SR 61.9, QR 43.1). This result suggests that both components are necessary.
>
> Our interpretation is that the two objectives play complementary roles. GRPO provides a strong mode-seeking signal that pushes the policy toward higher-reward regions, while the DMPO term provides a mode-covering regularization effect that prevents collapse and preserves reward diversity. In other words, GRPO improves reward maximization, and DMPO stabilizes that improvement by shaping the policy toward a broader reward-supporting distribution. We will include this $\lambda = \infty$ ablation in the appendix of the revised paper.
>
> **[4] Evaluation metrics for reasoning tasks**
>
> We apologize for the lack of clarity regarding the metrics used in Table 3. The table is **not** evaluated uniformly at pass@1.
>
> As stated in Appendix C, for mathematical reasoning benchmarks, we report **avg@32** for AIME 2024, AIME 2025, and AMC, while we use **pass@1** for Math500, OlympiadBench, and Minerva. Therefore, our current evaluation already covers both single-sample and multi-sample settings.
>
> Importantly, DMPO improves performance on both the avg@32 benchmarks and the pass@1 benchmarks. We believe this provides strong evidence that the method improves the overall reasoning distribution, rather than merely over-optimizing a single greedy mode.
>
> **References**
>
> [1] Chen, Zhipeng, et al. *Pass@k Training for Adaptively Balancing Exploration and Exploitation of Large Reasoning Models.* arXiv preprint arXiv:2508.10751, 2025.
>
> [2] Tang, Yunhao, et al. *Optimizing Language Models for Inference-Time Objectives Using Reinforcement Learning.* arXiv preprint arXiv:2503.19595, 2025.
>
> [3] Zhu, Xuekai, et al. *FlowRL: Matching Reward Distributions for LLM Reasoning.* arXiv preprint arXiv:2509.15207

---

> > ### Author Rebuttal · Reviewer_NRHx · 2026-04-02
> >
> > The authors have addressed my key questions. I will increase my score.

---

> > > ### Author Response · Authors · 2026-04-07
> > >
> > > This is a kind reminder regarding the review score. Since you mentioned that you would consider raising the score after the concerns are addressed, we would greatly appreciate it if you could kindly update the score when convenient. Thank you for your time and consideration.

---

### Decision · Program_Chairs · 2026-04-30

**Decision:**

Accept (regular)

**Comment:**

The reviewers unanimously agree that this paper addresses a critical bottleneck in the training of large reasoning models—mode collapse under on-policy reinforcement learning—by introducing a well-motivated solution (DMPO). The proposed approach of approximating forward KL minimization at the group level to encourage mode-covering behavior is conceptually sound and simple to integrate with standard methods like GRPO. Furthermore, the empirical results demonstrate improvements in both the quality and diversity of solutions across combinatorial optimization (NP-Bench) and mathematical reasoning tasks. During the rebuttal phase, the authors resolved the reviewers' primary concerns by giving additional comparisons against pass@k and GFlowNet baselines, clarifying the theoretical nuances linking their MSE proxy to forward KL minimization, and expanding the discussion on the method's limitations in extreme sparse-reward settings. Because the paper is technically solid and offers a useful contribution to the reasoning and reinforcement learning communities, I recommend acceptance.